# Simple Calibration via Geodesic Kernels

**Jayanta Dey**                                                                        *jdey4@jhmi.edu*
*Department of Biomedical Engineering*
*Johns Hopkins University*

**Haoyin Xu** *                                                                           *hx@jhu.edu*
*Department of Biomedical Engineering*
*Johns Hopkins University*

**Ashwin De Silva** *                                                                  *ldesilv2@jhu.edu*
*Department of Biomedical Engineering*
*Johns Hopkins University*

**Joshua T. Vogelstein**                                                              *jovo@jhu.edu*
*Department of Biomedical Engineering*
*Johns Hopkins University*

**Reviewed on OpenReview:** *https://openreview.net/forum?id=dpcRp8ix5T*

## Abstract

Deep discriminative approaches, such as decision forests and deep neural networks, have recently found applications in many important real-world scenarios. However, deploying these learning algorithms in safety-critical applications raises concerns, particularly when it comes to ensuring calibration for both in-distribution and out-of-distribution regions. Many popular methods for in-distribution (ID) calibration, such as isotonic and Platt's sigmoidal regression, exhibit adequate ID calibration performance. However, these methods are not calibrated for the entire feature space, leading to overconfidence in the out-of-distribution (OOD) region. Existing OOD calibration methods generally exhibit poor ID calibration. In this paper, we jointly address the ID and OOD problems. We leveraged the fact that deep models learn to partition feature space into a union of polytopes, that is, flat-sided geometric objects. We introduce a geodesic distance to measure the distance between these polytopes and further distinguish samples within the same polytope using a Gaussian kernel. Our experiments on both tabular and vision benchmarks show that the proposed approaches, namely Kernel Density Forest (`KDF`) and Kernel Density Network (`KDN`), obtain well-calibrated posteriors for both ID and OOD samples, while mostly preserving the classification accuracy and extrapolating beyond the training data to handle OOD inputs appropriately.

## 1 Introduction

Machine learning methods, specifically deep neural networks and decision forests (such as random forests and gradient boosting trees), show excellent performance in many real-world tasks (Xu et al., 2021a), including drug discovery (Cano et al., 2017; Shi et al., 2019), autonomous driving (Yurtsever et al., 2020), and clinical surgery (Bihorac et al., 2019). However, Calibrating confidence across the entire feature space for these approaches continues to be a significant challenge (Hein et al., 2019), as most existing algorithms focus either on in-distribution (ID) calibration or out-of-distribution (OOD) detection, rather than treating them as a unified problem. Calibrated confidence within the training or in-distribution (ID) region as well as in the out-of-distribution (OOD) region is crucial for safety critical applications like autonomous driving and

---

*Equal contribution.

computer-assisted surgery, where any aberrant reading should be detected and taken care of immediately (Hein et al., 2019; Meinke et al., 2021).

The methods for detecting out-of-distribution (OOD) samples in the literature can be broadly categorized into two types: discriminative and generative approaches. A straightforward strategy for OOD detection is to design a function that assigns higher scores to in-distribution (ID) samples and lower scores to OOD samples (Liang et al., 2017). Discriminative approaches typically modify the loss function (Nandy et al., 2020; Wan et al., 2018; DeVries & Taylor, 2018) or extensively train the model on OOD datasets to produce these distinct scores (Hendrycks et al., 2018; Hein et al., 2019). Recently, Hein et al. (2019) showed `ReLU` networks produce arbitrarily high confidence as the inference point moves far away from the training data and it prohibits most algorithms from having an asymptotic performance guarantee. Therefore, calibrating `ReLU` networks for the whole OOD region is not possible without fundamentally changing the network architecture. Others learn generative models for the ID and the OOD samples. The general idea is to perform the likelihood ratio test for a particular sample using generative models Ren et al. (2019), or to threshold the ID likelihoods to detect OOD samples. However, it is not obvious how to control the likelihoods far away from the training data for powerful generative models such as variational autoencoders (VAE) (Kingma et al., 2019) and adversarial generative networks (GAN) (Goodfellow et al., 2020). Moreover, Nalisnick et al. (2018) and Hendrycks et al. (2018) showed that VAEs and GANs can yield overconfident likelihoods far away from the training data. Although these algorithms focus solely on OOD detection for deep networks, other methods, such as those that partition the feature space (e.g. decision forests), can also exhibit overconfidence in OOD regions.

The algorithms described above are concerned only with OOD detection and do not address confidence calibration within the ID region at all. This is addressed by a separate group of algorithms (Zadrozny & Elkan, 2001; Caruana, 2004; Platt et al., 1999; Guo et al., 2017; 2019; Kull et al., 2019) that deals with ID confidence calibration.

We estimate posteriors that are calibrated in both ID and OOD regions. Our approach subsumes OOD detection and ID calibration by addressing the calibration problem as a continuum from ID to OOD regions (see Appendix Section A for further clarification). We conceptualize decision forests and `ReLU` networks as partitioning the input space into unions of polytopes, where each polytope has a distinct affine transformation. Nearby polytopes have similar affine functions. We used a "Geodesic" kernel to measure distances between these polytopes (Madhyastha et al., 2020), allowing us to better capture their relationships. We find the nearest polytope to an inference point and replace the affine function learned over that polytope with a Gaussian kernel, leading to two novel calibration methods: Kernel Density Forest (`KDF`) and Kernel Density Network (`KDN`). Our approach eliminates the need for training on OOD examples, enabling fully unsupervised OOD calibration. Through extensive simulations and benchmark experiments, we show that KDF and KDN achieve well-calibrated predictions for OOD samples while maintaining good accuracy and calibration in the ID region, on both tabular and vision benchmarks.

## 2 Methods

### 2.1 Setting

Consider a supervised learning problem with independent and identically distributed training samples $\{(\mathbf{x}_i, y_i)\}_{i=1}^n$ such that $(\mathbf{X}, Y) \sim P_{X,Y}$, where $\mathbf{X} \sim P_X$ is a $\mathcal{X} \subseteq \mathbb{R}^D$ valued input and $Y \sim P_Y$ is a $\mathcal{Y} = \{1, \cdots, K\}$ valued class label. Let $\mathcal{S}$ be the high density region of the marginal, $P_X$, thus $\mathcal{S} \subsetneq \mathcal{X}$. The posterior probability for class $y$ is given by the Bayes formula:

$$P_{Y|X}(y|\mathbf{x}) = \frac{P_{X|Y}(\mathbf{x}|y)P_Y(y)}{\sum_{k=1}^K P_{X|Y}(\mathbf{x}|k)P_Y(k)}, \quad \forall y \in \mathcal{Y}. \tag{1}$$

Here $P_{X|Y}(\mathbf{x}|y)$ is the class conditional density. In the OOD region where $\mathbf{x} \notin \mathcal{S}$, we do not have any evidence to favor a particular class $y$ for an inference point $\mathbf{x}$. Hence we can write:

$$P_{X|Y}(\mathbf{x}|y_i) = P_{X|Y}(\mathbf{x}|y_j), \forall y_i, y_j \in \mathcal{Y}. \tag{2}$$

We will refer to $P_{X|Y}(\mathbf{x}|y)$ as $f_y(\mathbf{x})$ hereafter for brevity. Substituting Equation 2 into Equation 1 for a data point in the out-of-distribution region yields:

$$P_{Y|X}(y|\mathbf{x}) = \frac{P_Y(y)}{\sum_{k=1}^{K} P_Y(k)} = P_Y(y). \tag{3}$$

Our goal is to learn a confidence score, $\mathbf{g} : \mathbb{R}^D \to [0,1]^K$, $\mathbf{g}(\mathbf{x}) = [g_1(\mathbf{x}), g_2(\mathbf{x}), \ldots, g_K(\mathbf{x})]$ such that,

$$g_y(\mathbf{x}) = \begin{cases} P_{Y|X}(y|\mathbf{x}), & \text{if } \mathbf{x} \in \mathcal{S} \\ P_Y(y), & \text{if } \mathbf{x} \notin \mathcal{S} \end{cases}, \quad \forall y \in \mathcal{Y} \tag{4}$$

## 2.2 Main Idea

Deep discriminative networks partition the feature space $\mathbb{R}^d$ into a union of $p$ affine polytopes $Q_r$ such that $\bigcup_{r=1}^{p} Q_r = \mathbb{R}^d$, and learn an affine function over each polytope (Hein et al., 2019; Xu et al., 2021b; Black et al., 2022; Priebe et al., 2020). Mathematically, the unnormalized class-conditional density for the label $y$ estimated by these deep discriminative models at a particular point $\mathbf{x}$ can be expressed as:

$$\hat{f}_y(\mathbf{x}) = \sum_{r=1}^{p} (\mathbf{a}_r^\top \mathbf{x} + b_r) \mathbb{1}(\mathbf{x} \in Q_r). \tag{5}$$

For example, in the case of a decision tree, $\mathbf{a}_r = \mathbf{0}$, i.e., decision trees assume a uniform distribution for the class-conditional densities within a leaf node. Among these polytopes, the ones that lie on the boundary of the training data extend to the whole feature space and hence encompass all over the OOD region. Since the posterior probability for a class is determined by the affine activation over each of these polytopes, the algorithms tend to be overconfident when making predictions on the OOD inputs. Moreover, there exist some polytopes that are not populated with training data. These unpopulated polytopes serve to interpolate between the training sample points. If we replace the affine activation function of the polytope with a Gaussian kernel and prune the polytopes unpopulated by the training data, the tail of the kernel will help interpolate between the training sample points while assigning lower likelihood to the low density or unpopulated polytope regions of the feature space.

## 2.3 Proposed Approach

We will call the above discriminative approaches as the 'parent approach' hereafter. Consider the collection of polytope indices $\mathcal{P}$ from the parent approach which are populated by the training data. We replace the affine functions over the populated polytopes with Gaussian kernels $\mathcal{G}(\cdot; \hat{\mu}_r, \hat{\Sigma}_r)$. For a particular inference point $\mathbf{x}$, we consider the Gaussian kernel with the minimum distance from the center of the kernel to the corresponding point:

$$\hat{r}_\mathbf{x}^* = \operatorname*{argmin}_r \|\mu_r - \mathbf{x}\|, \tag{6}$$

where $\|x_1 - x_2\|$ denotes a distance between $x_1$ and $x_2$. As we will show later, the type of distance considered in Equation 6 highly impacts the performance of the proposed model. In short, we modify Equation 5 from the parent ReLU-net or decision forest to estimate the class-conditional density:

$$\tilde{f}_y(\mathbf{x}) = \frac{1}{n_y} \sum_{r \in \mathcal{P}} n_{ry} \mathcal{G}(\mathbf{x}; \hat{\mu}_r, \hat{\Sigma}_r) \mathbb{1}(r = \hat{r}_\mathbf{x}^*), \tag{7}$$

where $n_y$ is the total number of training samples with label $y$ and $n_{ry}$ is the number of training samples from class $y$ that end up in polytope $Q_r$. Note that our proposed model is not a Gaussian mixture like model; rather, for a given inference point, it considers only the single nearest Gaussian. We add a small constant to the class conditional density so that the class conditional likelihoods become constant far away from the Gaussian center:

$$\hat{f}_y(\mathbf{x}) = \tilde{f}_y(\mathbf{x}) + \frac{b}{\log(n)}. \tag{8}$$

Note that in Equation 8, $\frac{b}{\log(n)} \to 0$ as the total training points, $n \to \infty$. See the derivation of Proposition 2 in Appendix B for further clarification on how the above constant helps in achieving our goal in Equation 4. The confidence score $\hat{g}_y(\mathbf{x})$ for class $y$ given a test point $\mathbf{x}$ is estimated using the Bayes rule as:

$$\hat{g}_y(\mathbf{x}) = \frac{\hat{f}_y(\mathbf{x})\hat{P}_Y(y)}{\sum_{k=1}^{K} \hat{f}_k(\mathbf{x})\hat{P}_Y(k)}, \tag{9}$$

where $\hat{P}_Y(y)$ is the empirical prior probability of class $y$ estimated from the training data. We obtain the class for a particular inference point $\mathbf{x}$ by:

$$\hat{y} = \underset{y \in \mathcal{Y}}{\operatorname{argmax}} \, \hat{g}_y(\mathbf{x}). \tag{10}$$

### 2.4 Geodesic Kernel

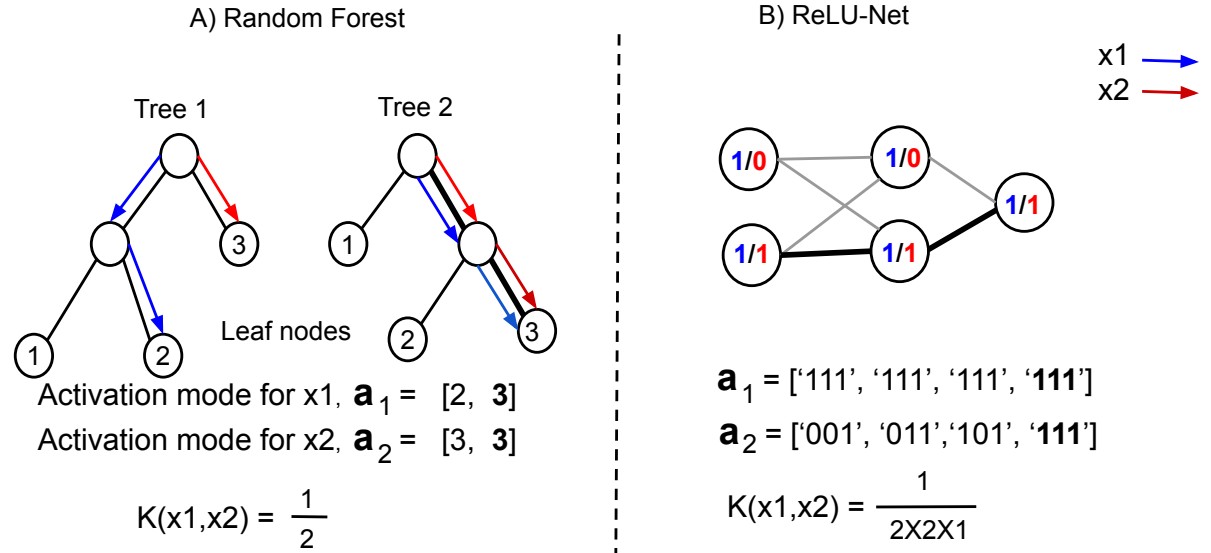

Figure 1: **Geodesic kernel calculation for two samples in decision forest and ReLU-net.** Two samples $\mathbf{x}_1$ (marked blue) and $\mathbf{x}_2$ (marked red) are pushed down A) a decision forest with two trees. B) a ReLU-net with three layers. The ReLU activation for the two samples at each node is shown with colors blue and red. The activation paths that are identically activated for the two samples above are bolded, along with the common elements in their activation modes.

As evident in Equation 5, samples that fall within the same polytope undergo the same affine transformation by the deep network. Black et al. (2022) identified polytopes as the fundamental building blocks of deep networks, introducing the concept of a "spline code," which represents the binarized and flattened activation pattern of the network for an input sample. These spline codes uniquely identify the polytope in which a sample falls and, as demonstrated by Black et al. (2022), can effectively cluster identical samples without even considering the absolute magnitude of activations. They further proposed using the Hamming distance between spline codes to identify nearby polytopes undergoing nearly similar transformations to a certain polytope in the network. Moreover, various other kernels based on activation patterns in deep networks have been explored in the literature (Balestriero et al., 2018; Wilson et al., 2016). However, to our knowledge, most of these kernels either do not construct a metric space or at most construct a semi-metric space and hence, they cannot be used as valid metrics (Burago, 2001). In the following, we propose a generalized kernel for both the decision forest and ReLU-net, and, as we will show in Proposition 1, it constructs a metric

space. This enables us to use our kernel to measure the distance between samples in the representation space learned by the network.

Our proposed kernels are based on existing random partition kernels (Davies & Ghahramani, 2014). Hence, all the properties of such partition kernels hold for our proposed kernel. The only difference is that—- we do not use random partitions of the feature space, instead we use the partitions learned by decision forests or ReLU-net after being trained on the training data. We identify these partitions using the leaf identities in decision forest or the activation patterns in a ReLU-net. Note that the geometry of these partitions or polytopes encodes the low-dimensional structure learned by the model (Balestriero et al., 2018).

### 2.4.1 Forest Kernel

A decision forest with $B$ decision trees partitions the feature space into a set of disjoint polytopes, $\{Q_r\}_{r=1}^p$. Decision forests map any sample $\boldsymbol{x}_i \in Q_r$ to a $B$ dimensional vector $\boldsymbol{a}_i$ where each element $a_i(t)$ of the vector represents the leaf identities the sample falls in each tree. We call the above vector $\boldsymbol{a}_i$ the "activation mode".

*If the two activation modes for two different training samples are identical, they belong to the same polytope.* In other words, if $\boldsymbol{a}_i = \boldsymbol{a}_j$, then $Q_r = Q_s$. This statement holds because the above samples will end up in the same partition in all decision trees.

For any two points $\mathbf{x}_i \in Q_r$ and $\mathbf{x}_j \in Q_s$, we define the kernel $\mathcal{K}(\boldsymbol{x}_i, \boldsymbol{x}_j)$ as:

$$\mathcal{K}(\boldsymbol{x}_i, \boldsymbol{x}_j) = \frac{1}{B} \sum_{t=1}^{B} \mathbb{1}(a_i(t) = a_j(t)). \tag{11}$$

Note that $0 \leq \mathcal{K}(\boldsymbol{x}_i, \boldsymbol{x}_j) \leq 1$. In short, $\mathcal{K}(\boldsymbol{x}_i, \boldsymbol{x}_j)$ is the fraction of total trees where the two samples follow the same path from the root to a leaf node.

### 2.4.2 Network Kernel

A fully connected $L$ layer ReLU-net partitions the feature space into a set of disjoint polytopes, $\{Q_r\}_{r=1}^p$. We have the set of all nodes in a particular layer $l$ denoted by $\mathcal{N}_l = \{n_l^i\}_{i=1}^{|\mathcal{N}_l|}$. We can randomly pick a node $n_l^{i_l} \in \mathcal{N}_l$ at each layer $l$, and construct a sequence of nodes starting at the input layer and ending at the output layer, which we call an **activation path**: $m = \{n_l^{i_l} \in \mathcal{N}_l\}_{l=1}^L$. Note that there are $B = \Pi_{l=1}^L |\mathcal{N}_l|$ possible activation paths for a sample in the ReLU-net. We index each path by a unique identifier number $t \in \{1, \cdots, B\}$.

While pushing a training sample $\mathbf{x}_i \in Q_r$ through the network, we define the activation from a ReLU unit at any node as '1' when it has positive output and '0' otherwise. Therefore, the activation indicates on which side of the affine function at each node the sample falls. The activation of all nodes on an activation path $m_t$ for a particular sample creates a string of binary numbers $a_i(t) \in \{0, 1\}^L$. Therefore, a ReLU-net maps any sample $\boldsymbol{x}_i \in Q_r$ to a $B$ dimensional vector $\boldsymbol{a}_i$ where each element $a_i(t)$ of the vector represents a string of binary numbers of length $L$. We call the above vector $\boldsymbol{a}_i$ the "activation mode". Note that similar to the leaf identities in decision forests, each element $a_i(t)$ of the above vector represents a partition of the feature space, that is, only the samples belonging to a particular region of the feature space can have $a_i(t)$. The intersection of all such overlapping partitions in an activation mode gives the polytope $Q_r$ in which the sample falls.

*If the two activation modes for two different training samples are identical, they belong to the same polytope.* In other words, if $\boldsymbol{a}_i = \boldsymbol{a}_j$, then $Q_r = Q_s$. This statement holds because the above samples will lie on the same side of the affine function at each node in different layers of the network. Now, we define the kernel $\mathcal{K}(\boldsymbol{x}_i, \boldsymbol{x}_j)$ as:

$$\mathcal{K}(\boldsymbol{x}_i, \boldsymbol{x}_j) = \frac{1}{B} \sum_{t=1}^{B} \mathbb{1}(a_i(t) = a_j(t)). \tag{12}$$

Note that $0 \leq \mathcal{K}(\boldsymbol{x}_i, \boldsymbol{x}_j) \leq 1$. In short, $\mathcal{K}(\boldsymbol{x}_i, \boldsymbol{x}_j)$ is the fraction of total activation paths that are activated identically for two samples in two different polytopes $r$ and $s$. A generalized pseudocode outlining the above

two kernel calculation is provided in Algorithm 1. See Figure 1 for a visual interpretation of the kernel calculation.

---

**Algorithm 1** Computing Geodesic Kernel

---

**Input:**
    (1) $\mathbf{x}_i, \mathbf{x}_j \in \mathbb{R}^{1 \times d}$                                                      ▷ two input samples
    (2) $\theta$                    ▷ parameters for parent decision forest or `ReLU`-net
**Output:** $\mathcal{K}(\boldsymbol{x}_i, \boldsymbol{x}_j) \in [0, 1]$                  ▷ compute similarity between $i$ and $j$-th samples.
 1: **function** COMPUTEKERNEL($\mathbf{x}_i, \mathbf{x}_j, \theta$)
 2:     $\boldsymbol{a}_i \leftarrow$ PUSHDOWNNETWORK($\mathbf{x}_i, \theta$)        ▷ push samples down the deep network to get the activation modes.
 3:     $\boldsymbol{a}_j \leftarrow$ PUSHDOWNNETWORK($\mathbf{x}_j, \theta$)
 4:     $M \leftarrow$ COUNTMATCHES($\boldsymbol{a}_i, \boldsymbol{a}_j$)     ▷ count the number of similar activation modes or leaf identities
 5:     $\mathcal{K}(\boldsymbol{x}_i, \boldsymbol{x}_j) \leftarrow \frac{M}{B}$                        ▷ $B$ is the dimension of activation modes.
 6:     **return** $\mathcal{K}(\boldsymbol{x}_i, \boldsymbol{x}_j)$
 7: **end function**

---

## 2.5 Network Kernel with Deep Network

In this section, we empirically assess the validity of the kernel defined in Equation 12 as network depth increases. We conduct experiments on a two-dimensional Gaussian XOR simulation (described in Appendix D) and the 784-dimensional Fashion-MNIST dataset (from OpenML-CC18 (Bischl et al., 2017)) using fully-connected networks of varying depth and width. Specifically, we measure the average kernel similarity between sample pairs from the same and different class labels, both before training (with random initialization) and after training. As shown in Figure 2, kernel similarity decreases monotonically with depth under random initialization, indicating increasingly disjoint activation paths. In contrast, after training, similarity between same-class samples remains relatively high, suggesting that learning encourages alignment in activation paths. These results demonstrate that, while the kernel becomes sparse in untrained deep networks with random weights, it retains its discriminative power once the network is trained.

## 2.6 Geodesic Distance

Consider $\mathcal{P}_n = \{Q_1, Q_2, \cdots, Q_p\}$ as a partition of $\mathbb{R}^d$ given by a decision forest or a `ReLU`-net after being trained on $n$ training samples. We measure distance between two points $\mathbf{x}_i \in Q_r, \mathbf{x}_j \in Q_s$ using the kernel introduced in Equation 11 and Equation 12, and call it 'Geodesic' distance (Schölkopf, 2000):

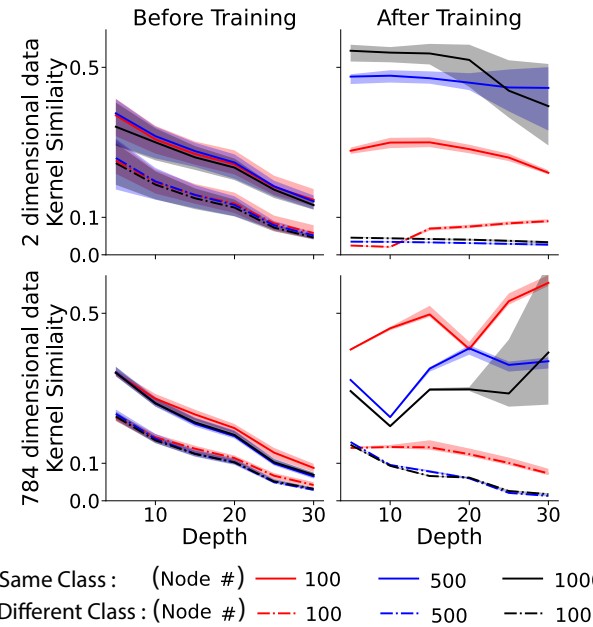

Figure 2: **Average kernel values are higher for similar-class pairs than for dissimilar ones.** Values decrease with depth in networks with random partition (left) but remain stable when trained on data (right).

$$d(\boldsymbol{x}_i, \boldsymbol{x}_j) = -\mathcal{K}(\boldsymbol{x}_i, \boldsymbol{x}_j) + \frac{1}{2}(\mathcal{K}(\boldsymbol{x}_i, \boldsymbol{x}_i) + \mathcal{K}(\boldsymbol{x}_j, \boldsymbol{x}_j)) = 1 - \mathcal{K}(\boldsymbol{x}_i, \boldsymbol{x}_j). \tag{13}$$

**Proposition 1.** $(\mathcal{P}_n, d)$ *is a metric space.*

*Proof.* See Appendix B.1 for the proof. □

The above proposition enables us to use Geodesic distance as a metric in our algorithm. We use Geodesic distance in Equation 6 to find the nearest polytope to the inference point. As Geodesic distance cannot distinguish between points within the same polytope, it has a resolution similar to the size of the polytope. We assume the polytope is small enough so that the manifold within a polytope can be considered Euclidean and hence for discriminating between two points within the same polytope, we fit a Gaussian kernel within the polytope (described below). Moreover, Gaussian kernels decay exponentially from their center which improves our OOD calibration. In Section 3, we will empirically show that using Geodesic distance scales better with higher dimensions compared to Euclidean distance.

## 2.7 Gaussian Kernel Parameter Estimation

We fit Gaussian kernel parameters to the samples that end up in the $r$-th polytope. We set the kernel center along the $d$-th dimension:

$$\hat{\mu}_r^d = \frac{1}{n_r} \sum_{i=1}^{n} x_i^d \mathbb{1}(\mathbf{x}_i \in Q_r), \tag{14}$$

where $x_i^d$ is the value of $\mathbf{x}_i$ along the $d$-th dimension. We set the kernel variance along the $d$-th dimension:

$$(\hat{\sigma}_r^d)^2 = \frac{1}{n_r} \{ \sum_{i=1}^{n} \mathbb{1}(\mathbf{x}_i \in Q_r)(x_i^d - \hat{\mu}_r^d)^2 + \lambda \}, \tag{15}$$

where $\lambda$ is a small constant that prevents $\hat{\sigma}_r^d$ from being 0. We constrain our estimated Gaussian kernels to have diagonal covariance.

## 2.8 Sample Size Ratio Estimation

For a high dimensional dataset with low training sample size, the polytopes are sparsely populated with training samples. For improving the estimate of the ratio $\frac{n_{ry}}{n_y}$ in Equation 7, we incorporate the samples from other polytopes $Q_s$ based on the similarity $w_{rs}$ between $Q_r$ and $Q_s$ as:

$$\frac{\tilde{n}_{ry}}{\tilde{n}_y} = \frac{\sum_{s \in \mathcal{P}} \sum_{i=1}^{n} w_{rs} \mathbb{1}(\mathbf{x}_i \in Q_s) \mathbb{1}(y_i = y)}{\sum_{r \in \mathcal{P}} \sum_{s \in \mathcal{P}} \sum_{i=1}^{n} w_{rs} \mathbb{1}(\mathbf{x}_i \in Q_s) \mathbb{1}(y_i = y)}. \tag{16}$$

As $n \to \infty$, the estimated weights $w_{rs}$ should satisfy the condition:

$$w_{rs} \to \begin{cases} 0, & \text{if } Q_r \neq Q_s \\ 1, & \text{if } Q_r = Q_s. \end{cases} \tag{17}$$

Note that if we satisfy the conditions in Equation 17, then we have $\frac{\tilde{n}_{ry}}{\tilde{n}_y} \to \frac{n_{ry}}{n_y}$ as $n \to \infty$. We exponentiate $\mathcal{K}(r, s)$ so that the conditions in Equation 17 is satisfied:

$$w_{rs} = \mathcal{K}(\boldsymbol{x}_i, \boldsymbol{x}_j)^{\gamma \log n}. \tag{18}$$

We choose $\gamma$ using grid search on a hold-out dataset. Therefore, we modify Equation 7 as:

$$\hat{f}_y(\mathbf{x}) = \frac{1}{\tilde{n}_y} \sum_{r \in \mathcal{P}} \tilde{n}_{ry} \mathcal{G}(\mathbf{x}; \hat{\mu}_r, \hat{\Sigma}_r) \mathbb{1}(r = \hat{r}_{\mathbf{x}}^*), \tag{19}$$

where $\hat{r}_{\mathbf{x}}^* = \text{argmin}_r \|\hat{\mu}_r - \mathbf{x}\|$. Now we use $\hat{f}_y(\mathbf{x})$ estimated using (19) in Equation (8), (9) and (10), respectively.

Given $n$ training samples $\{(\mathbf{x}_i, y_i)\}_{i=1}^{n}$, we define the distance of an inference point $\mathbf{x}$ from the training points as: $d_{\mathbf{x}} = \min_{i=1,\cdots,n} \|\mathbf{x} - \mathbf{x}_i\|$, where $\|\cdot\|$ denotes Euclidean distance.

**Proposition 2** (Asymptotic OOD Convergence)**.** *Given non-zero and bounded bandwidth of the Gaussians, then we have:*

$$\lim_{d_{\mathbf{x}} \to \infty} \hat{g}_y(\mathbf{x}) = \hat{P}_Y(y).$$

*Proof.* See Appendix B.2 for the proof. □

Note that the above proposition guarantees asymptotic performance of the proposed approach. In contrast, typical existing approaches lack any such guarantees. Below we conduct empirical experiments including simulations and benchmark datasets to gain insights about the non-asymptotic performance of the proposed approach.

---

**Algorithm 2** Fit a `KDX` model.

---
**Input:**
    (1) $\theta$                       ▷ Parent learner (random forest or deep network model)
    (2) $\mathcal{D}_n = (\mathbf{X}, \mathbf{y}) \in \mathbb{R}^{n \times d} \times \{1, \dots, K\}^n$          ▷ Training data
**Output:** $\mathcal{G}$                            ▷ a KDX model
 1: **function** KDX.FIT$(\theta, \mathbf{X}, \mathbf{y})$
 2:     **for** $i = 1, \dots, n$ **do**             ▷ Iterate over the dataset to calculate the weights
 3:         **for** $j = 1, \dots, n$ **do**
 4:             $k_{ij} \leftarrow$ COMPUTEKERNEL$(\mathbf{x}_i, \mathbf{x}_j, \theta)$
 5:         **end for**
 6:     **end for**
 7:
 8:     $\{\{k_{rs}\}_{r=1}^{\tilde{p}}\}_{s=1}^{\tilde{p}} \leftarrow$ GETPOLYTOPES$(\mathbf{k})$    ▷ Identify the polytopes populated by the training data by merging two samples with $k_{ij} = 1$
 9:
10:     **for** $r = 1, \dots, \tilde{p}$ **do**                     ▷ Iterate over each polytope
11:         $\mathcal{G}.\hat{\mu}_r, \mathcal{G}.\hat{\Sigma}_r, \mathcal{G}.\hat{n}_{ry} \leftarrow$ ESTIMATEPARAMETERS$(\mathbf{X}, y, \{k_{rs}\}_{s=1}^{\tilde{p}})$   ▷ Estimate model parameters
12:     **end for**
13:     **return** $\mathcal{G}$
14: **end function**

---

## 2.9 Performance Measure

For the simulation setups, we evaluate performance with classification error, Hellinger distance (Kailath, 1967; Rao, 1995) between the true and estimated class conditional posteriors, and mean max confidence (MMC) (Hein et al., 2019). We use expected calibration error (ECE) to evaluate in-distribution calibration for the OpenML-CC18 data suite, with $R = 20$ bins for all datasets Guo et al. (2017). Given $n$ OOD samples $\{\mathbf{x}_i\}_{i=1}^n$, we define OOD calibration error (OCE) to measure OOD performance for the benchmark datasets as:

$$\text{OCE} = \frac{1}{n} \sum_{i=1}^{n} \left| \max_{y \in \mathcal{Y}} (\hat{P}_{Y|X}(y|\mathbf{x}_i)) - \max_{y \in \mathcal{Y}} (P_Y(y)) \right|. \tag{20}$$

Note that one needs to know the true prior to calculating OCE. For tabular data experiments, training points are sampled based on empirical class priors in the overall dataset. In vision experiments, we assume equal class priors and sample training points accordingly.

## 3 Empirical Results

For empirical experiments, we first demonstrate that our methods satisfy the theoretical guarantees on simulated datasets where we know the ground truth, and then we apply our approach on benchmark datasets

such as OpenML-CC18 (Bischl et al., 2017) [1] and different vision datasets. The details of the simulation data sets and hyperparameters used for all experiments are provided in Appendix D. For tabular and vision data sets, we have used ID calibration approaches, such as ISOTONIC Zadrozny & Elkan (2001); Caruana (2004) and SIGMOID regression Platt et al. (1999), as baselines. Additionally, for the vision benchmark dataset, we provide results with OOD calibration approaches such as: ACET Hein et al. (2019), ODIN Liang et al. (2017), OE (outlier exposure) Hendrycks et al. (2018). For each approach, 70% of the training data was used to fit the model and the rest of the data was used to calibrate the model.

### 3.1 Empirical Study on Tabular Data

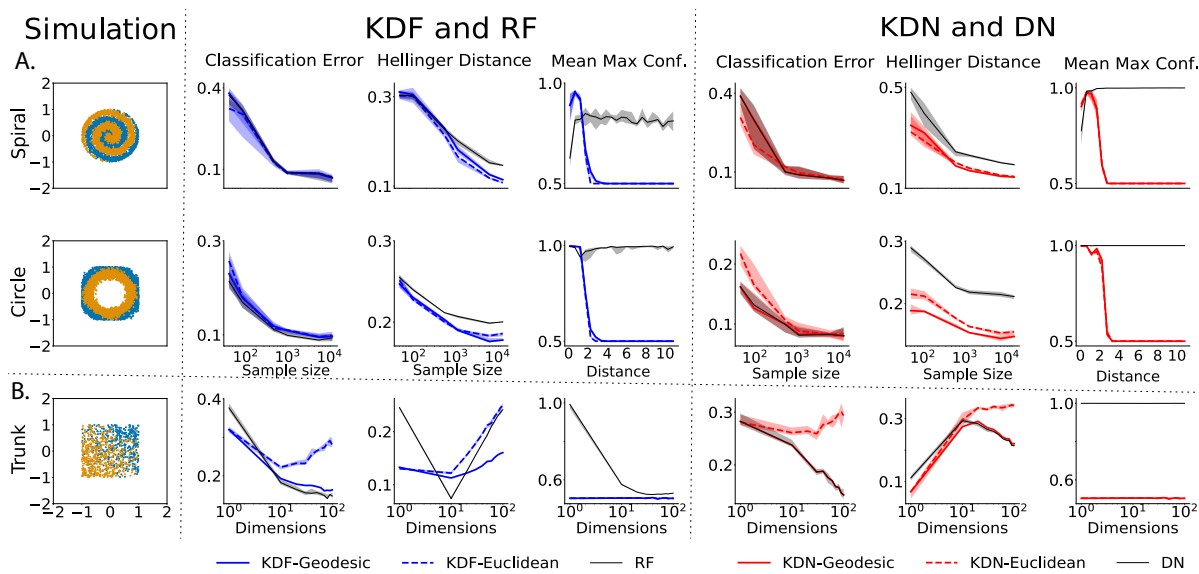

Figure 3: KDF and KDN **improve both in-distribution and out-of-distribution calibration of the parent model while preserving classification accuracy.** We evaluate performance using simulation datasets, reporting classification error, Hellinger distance from true posteriors, and mean max confidence (MMC). Results are presented for: A. Two-dimensional simulations. In the fourth and seventh columns, MMC is measured as a function of increasing distance from the origin and B. High-dimensional (Trunk) simulations — visualized along the first two dimensions. OOD points are randomly sampled from a sphere of radius 20, and MMC is measured across increasing dimensions. For Hellinger distance, lower values indicate a closer match to the true training distribution, while in OOD regions, a value near 0.5 signifies better OOD calibration. The median performance of 10 repetitions is shown as a dark curve with shaded region as error bars.

#### 3.1.1 Simulation Study

For simulation study on tabular data, we use 6 simulation datasets. Three of the simulations are visualized in Figure 3A and see Appendix D for additional simulations and details. We sample 10,000 training samples with half of the samples from each class. The samples are sampled within the region $[-1, 1] \times [-1, 1]$, rendering the empty annular region between $[-1, 1] \times [-1, 1]$ and $[-2, 2] \times [-2, 2]$ as the low density or OOD region. In the Trunk simulation (Figure 3 B), each of two classes is sampled from a Gaussian distribution, with higher dimensions having increasingly less discriminative information (Trunk, 1979). We use both Euclidean and Geodesic distances to detect the nearest polytope center. To measure in-distribution performance, we report classification error and the Hellinger distance between the estimated and true distributions. To measure OOD performance, we normalize the training data to the maximum of their norm $\ell_2$. For inference, we sample 1000 inference points uniformly from a circle and compute the mean max posterior for increasing

---

[1] https://www.openml.org/s/99

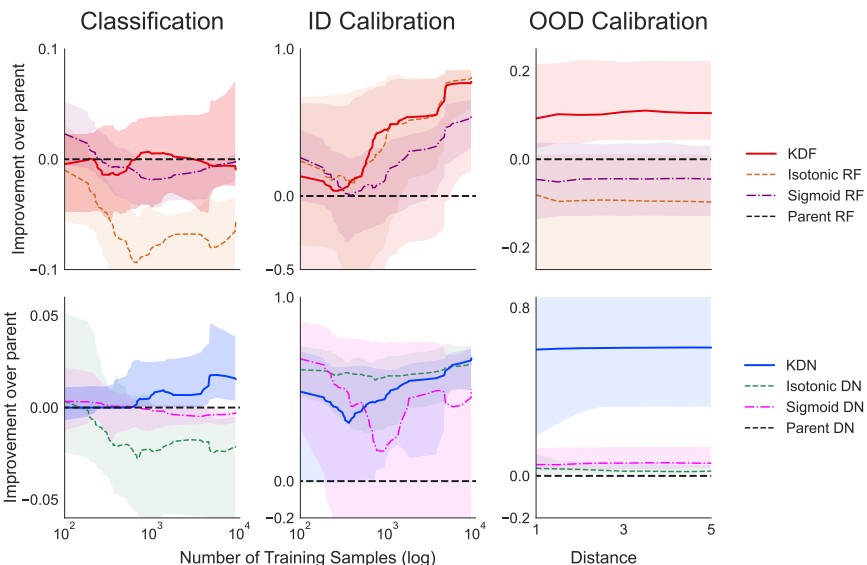

Figure 4: **Performance summary of KDF and KDN on OpenML-CC18 data suite.** The dark curve in the middle shows the median of performance on 45 datasets with the shaded region as error bar.

distance from the origin. Therefore, for a distance of up to 1 we have in-distribution samples and distances further than 1 are OOD. For the Trunk simulation, we randomly sampled OOD points from a sphere of fixed radius 20 and measured mean max confidence for increasing dimensions.

KDF and KDN achieve classification accuracy comparable to their parent algorithms (Figure 3A). However, they provide a more accurate estimation of the in-distribution (ID) region. The use of geodesic distance further enhances performance relative to Euclidean distance. As data points move farther from the training set, the mean maximum confidence for KDF and KDN approaches the maximum of the class priors, reaching 0.5. Furthermore, KDF-Geodesic and KDN-Geodesic exhibit better scalability in higher dimensional spaces compared to their Euclidean counterparts (Figure 3 B).

Hereafter, we only use Geodesic distance to find the nearest polytope in our algorithms.

### 3.1.2 OpenML-CC18 Data Study

We use OpenML-CC18 data suite for tabular benchmark dataset study. We exclude any dataset which contains categorical features or NaN values [2] and conduct our experiments on 45 datasets with varying dimensions and sample sizes. We repeat the experiment for each dataset 10 times. For the OOD experiments, we follow a similar setup as that of the simulation data. We normalize the training data by their maximum $\ell_2$ norm and sample 1000 testing samples uniformly from hyperspheres where each hypersphere has increasing radius starting from 1 to 5. For each dataset, we measure improvement with respect to the parent algorithm:

$$\frac{\mathcal{E}_p - \mathcal{E}_M}{\mathcal{E}_p}, \tag{21}$$

where $\mathcal{E}_p$ represents either classification error, ECE or OCE for the parent algorithm and $\mathcal{E}_M$ represents the performance of the approach in consideration. Positive improvement implies that the corresponding approach performs better than the parent approach. We report the median of improvement on different datasets along with the interquartile ranges (Figure 4, extended results for each dataset in the appendix). On average KDF and KDN have similar or better classification accuracy compared to their respective parent algorithms, whereas Isotonic and Sigmoid regression have lower classification accuracy most of the cases. KDF and KDN have similar ID calibration performance to the other baseline approaches, all better than the parent

---

[2]We also excluded the dataset with dataset id 23517 as we could not achieve better than chance accuracy using RF and DN on that dataset.

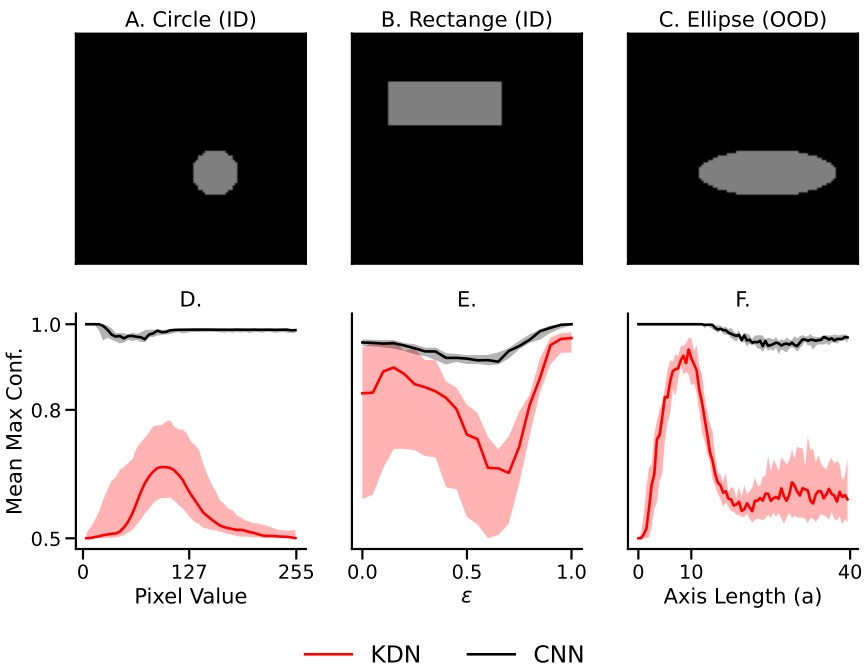

Figure 5: KDN **filters out inference points with different kinds of shifts from the training data.**
Simulated images: (A) circle with radius 10, (B) rectangle with sides $(20, 50)$ and (C) out-of-distribution
test points, ellipse with minor and major axis $(10, 30)$. Mean max confidence of KDN are plotted for semantic
shift of the inference points created by (D) changing the color intensity, (E) taking convex combination of
circle and rectangle, i.e., pixel-wise convex combination of the images in A and B, (F) changing one of the
axes of the ellipse.

algorithm for all sample sizes. Most interestingly, KDN and KDF improve OOD calibration of their respective
parent algorithms by a huge margin while the baseline approaches completely fail in OOD calibration.

### 3.2 Empirical Study on Vision Data

In vision data, each image pixel contains local information about the neighboring pixels. To extract the local
information, we use convolutional or vision transformer encoders at the front-end. Consider a set of images $\mathcal{I}$
where each image has width $W$, height $H$ and each pixel in an image is identified by $\{\mathbf{M}, w_1, w_2\}$, where $\boldsymbol{M}$
is a vector representing the RGB pixel values, $w_1$ is the horizontal location, $w_2$ is the vertical location. More
precisely, we have a front-end encoder, $h_e : \mathcal{I} \mapsto \mathbb{R}^d$ and typically, $d << W \times H$. After the encoder, there are
a few fully connected dense layers for discriminating among the $K$ class labels, $h_f : \mathbb{R}^d \mapsto \mathbb{R}^K$. Note that
the $d$-dimensional embedding outputs from the encoder are partitioned into polytopes by the dense layers
(see Equation (5)) and we fit a KDN on the embedding outputs.

#### 3.2.1 Simulation Study

For the simulation study, we use a CNN with one convolutional layer (kernel size $3 \times 3$) followed by two
fully connected layers with 10 and 2 nodes in each. Each experiment is repeated 10 times to get the error
bars. Consider two random variables $C_1, C_2 \sim U(0, 100)$. We sample 2000 circles $X_c$ and 2000 rectangles
$X_r$ grayscale training images using Equation 22 and 23 respectively. Each image has width 100 and height
100. The CNN is trained to classify an image into a circle or a rectangle.

$$M = \begin{cases} 127, & \text{if } (w_1 - C_1)^2 + (w_2 - C_2)^2 \leq 100 \\ 0, & \text{otherwise.} \end{cases} \tag{22}$$

Table 1: KDN **achieves good calibration at both ID and OOD regions whereas other approaches excel either in the ID or the OOD region. Notably, KDN has reduced confidence on wrongly classified ID points.** '↑' and '↓' indicate whether higher and lower values are better, respectively. OCE = OOD calibration error, MMC* = Mean Max Confidence on wrongly classified ID points.

(a) **In-Distribution (ID) Performance**

| ID dataset | Statistics | Parent | KDN | Isotonic | Sigmoid | ACET | ODIN | OE | Focal | textbfDual Focal |
|---|---|---|---|---|---|---|---|---|---|---|
| CIFAR-10 | Accuracy(%) ↑ | $98.06 \pm 0.00$ | $97.45 \pm 0.00$ | $98.16 \pm 0.00$ | $98.10 \pm 0.00$ | $\mathbf{98.23 \pm 0.00}$ | $97.97 \pm 0.00$ | $97.94 \pm 0.00$ | $94.79 \pm 0.00$ | $95.23 \pm 0.00$ |
|  | ECE ↓ | $0.01 \pm 0.00$ | $\mathbf{0.00 \pm 0.00}$ | $\mathbf{0.00 \pm 0.00}$ | $\mathbf{0.00 \pm 0.00}$ | $0.01 \pm 0.00$ | $0.02 \pm 0.00$ | $0.01 \pm 0.00$ | $0.07 \pm 0.00$ | $0.01 \pm 0.00$ |
|  | MMC* ↓ | $0.76 \pm 0.00$ | $\mathbf{0.65 \pm 0.08}$ | $0.74 \pm 0.02$ | $0.90 \pm 0.01$ | $0.86 \pm 0.02$ | $0.97 \pm 0.01$ | $0.69 \pm 0.01$ | $0.56 \pm 0.01$ | $0.69 \pm 0.01$ |
| CIFAR-100 | Accuracy(%) ↑ | $\mathbf{86.72 \pm 0.00}$ | $85.46 \pm 0.00$ | $85.33 \pm 0.00$ | $86.61 \pm 0.00$ | $85.07 \pm 0.00$ | $86.56 \pm 0.00$ | $86.09 \pm 0.00$ | $79.37 \pm 0.00$ | $80.81 \pm 0.00$ |
|  | ECE ↓ | $0.10 \pm 0.01$ | $\mathbf{0.02 \pm 0.00}$ | $0.04 \pm 0.01$ | $0.02 \pm 0.00$ | $0.11 \pm 0.01$ | $0.04 \pm 0.00$ | $0.03 \pm 0.00$ | $0.07 \pm 0.00$ | $0.02 \pm 0.00$ |
|  | MMC* ↓ | $0.46 \pm 0.02$ | $\mathbf{0.41 \pm 0.02}$ | $0.65 \pm 0.03$ | $0.45 \pm 0.02$ | $0.84 \pm 0.03$ | $0.77 \pm 0.01$ | $0.52 \pm 0.02$ | $0.46 \pm 0.00$ | $0.55 \pm 0.00$ |
| SVHN | Accuracy(%) ↑ | $93.52 \pm 0.01$ | $92.84 \pm 0.00$ | $93.64 \pm 0.01$ | $93.65 \pm 0.01$ | $93.22 \pm 0.01$ | $93.52 \pm 0.01$ | $\mathbf{93.76 \pm 0.01}$ | $60.39 \pm 0.00$ | $61.02 \pm 0.00$ |
|  | ECE ↓ | $0.01 \pm 0.00$ | $\mathbf{0.00 \pm 0.00}$ | $0.01 \pm 0.00$ | $0.06 \pm 0.00$ | $0.05 \pm 0.01$ | $0.06 \pm 0.01$ | $0.01 \pm 0.00$ | $0.06 \pm 0.00$ | $0.07 \pm 0.01$ |
|  | MMC* ↓ | $0.82 \pm 0.01$ | $0.63 \pm 0.00$ | $0.67 \pm 0.01$ | $0.78 \pm 0.00$ | $0.88 \pm 0.01$ | $0.97 \pm 0.01$ | $0.70 \pm 0.04$ | $0.33 \pm 0.00$ | $\mathbf{0.41 \pm 0.00}$ |

(b) **Out-of-Distribution (OOD) Performance (OCE ↓)**

| ID dataset | OOD dataset | Parent | KDN | Isotonic | Sigmoid | ACET | ODIN | OE | Focal | Dual Focal |
|---|---|---|---|---|---|---|---|---|---|---|
| CIFAR-10 | CIFAR-100 | $0.47 \pm 0.01$ | $\mathbf{0.12 \pm 0.01}$ | $0.47 \pm 0.01$ | $0.69 \pm 0.01$ | $0.57 \pm 0.01$ | $0.79 \pm 0.00$ | $0.29 \pm 0.01$ | $0.47 \pm 0.00$ | $0.51 \pm 0.00$ |
|  | SVHN | $0.87 \pm 0.00$ | $\mathbf{0.01 \pm 0.00}$ | $0.85 \pm 0.00$ | $0.69 \pm 0.01$ | $0.90 \pm 0.01$ | $0.87 \pm 0.00$ | $0.04 \pm 0.01$ | $0.40 \pm 0.04$ | $0.54 \pm 0.03$ |
|  | Noise | $0.28 \pm 0.08$ | $0.03 \pm 0.02$ | $0.30 \pm 0.04$ | $0.56 \pm 0.12$ | $\mathbf{0.01 \pm 0.00}$ | $0.53 \pm 0.09$ | $0.07 \pm 0.02$ | $0.52 \pm 0.11$ | $0.79 \pm 0.00$ |
| CIFAR-100 | CIFAR-10 | $0.23 \pm 0.01$ | $\mathbf{0.12 \pm 0.01}$ | $0.50 \pm 0.04$ | $0.22 \pm 0.01$ | $0.62 \pm 0.04$ | $0.54 \pm 0.01$ | $0.28 \pm 0.02$ | $0.31 \pm 0.00$ | $0.42 \pm 0.00$ |
|  | SVHN | $0.20 \pm 0.01$ | $\mathbf{0.12 \pm 0.01}$ | $0.47 \pm 0.06$ | $0.20 \pm 0.02$ | $0.58 \pm 0.05$ | $0.57 \pm 0.02$ | $0.20 \pm 0.02$ | $0.21 \pm 0.03$ | $0.30 \pm 0.03$ |
|  | Noise | $0.05 \pm 0.02$ | $0.03 \pm 0.00$ | $0.38 \pm 0.04$ | $0.07 \pm 0.06$ | $0.08 \pm 0.04$ | $0.18 \pm 0.06$ | $0.04 \pm 0.03$ | $0.04 \pm 0.03$ | $\mathbf{0.03 \pm 0.01}$ |
| SVHN | CIFAR-10 | $0.40 \pm 0.04$ | $\mathbf{0.14 \pm 0.01}$ | $0.37 \pm 0.02$ | $0.53 \pm 0.03$ | $0.67 \pm 0.04$ | $0.80 \pm 0.02$ | $\mathbf{0.14 \pm 0.01}$ | $0.31 \pm 0.00$ | $0.40 \pm 0.00$ |
|  | CIFAR-100 | $0.40 \pm 0.03$ | $\mathbf{0.14 \pm 0.01}$ | $0.37 \pm 0.03$ | $0.54 \pm 0.03$ | $0.66 \pm 0.03$ | $0.80 \pm 0.02$ | $0.16 \pm 0.01$ | $0.40 \pm 0.00$ | $0.48 \pm 0.00$ |
|  | Noise | $0.36 \pm 0.08$ | $0.12 \pm 0.04$ | $0.32 \pm 0.06$ | $0.48 \pm 0.06$ | $\mathbf{0.03 \pm 0.01}$ | $0.67 \pm 0.05$ | $0.15 \pm 0.03$ | $0.24 \pm 0.03$ | $0.32 \pm 0.04$ |

$$M = \begin{cases} 127, & \text{if } C_1 - 10 \leq w_1 < C_1 + 10 \\ & \text{and } C_2 - 25 \leq w_2 < C_2 + 25 \\ 0, & \text{otherwise.} \end{cases} \tag{23}$$

We perform three experiments while inducing different types of shifts in the inference points (Figure 5). In the first experiment, we sample inference points using Equation 22 and 23, but sweep over different intensities for the object (Figure 5 D). Therefore, the inference point is maximally ID for intensity at 127 and becomes more OOD as we move away from 127. In the second experiment, we kept the intensity at 127 while taking convex combination of a circle and a rectangle. Let images of circles and rectangles be denoted by $X_c$ and $X_r$. We derive an interference point as $X_{inf}$:

$$X_{inf} = \epsilon X_c + (1 - \epsilon) X_r. \tag{24}$$

Therefore, $X_{inf}$ is maximally distant from the training points for $\epsilon = 0.5$ and closest to the ID points at $\epsilon = \{0, 1\}$ (Figure 5 E).

In the third experiment, we sampled ellipse images using Equation 25, sweeping $a$ from 0.01 to 40 (Figure 5 F). The inference point becomes ID at $a = 10$ and remains OOD otherwise. As shown in Figure 5 (D, E, F), in all the experiments KDN becomes less confident for the OOD points while the parent CNN remains overconfident throughout different shifts of the test points.

$$M = \begin{cases} 127, & \text{if } \frac{(w_1 - C_1)^2}{a^2} + \frac{(w_2 - C_2)^2}{100} \leq 1 \\ 0, & \text{otherwise.} \end{cases} \tag{25}$$

### 3.2.2 Vision Benchmark Datasets Study

In this study, we use ViT_B16 (provided in keras-vit package) vision transformer encoders (Dosovitskiy et al., 2020) pretrained on ImageNet (Deng et al., 2009) dataset and fine-tuned on the ID datasets. As described in Section 3.2, we use a ViT_B16 encoder and use Equation 12 on the last two fully connected layers to calculate the geodesic kernel. We use the same encoder for all the baseline algorithms and fine-tune it

without freezing any weight. We experiment with popular benchmark datasets– CIFAR10, CIFAR-100 and SVHN. For each experiment, we train on one of the three datasets and test `KDN` for OOD performance on the other two datasets. We also consider a simple OOD dataset, noise, where we sample noise OOD samples of size $32 \times 32 \times 3$ according to a Uniform distribution with intensities within range $[0, 1]$. Although noise samples is easier to detect as OOD point, we observe that many of the baseline approaches perform poorly even on the noise samples. Each experiment is repeated for 5 different seeds to get an error bar. Table 1 shows that pretrained vision transformers are already well-calibrated for ID, and the OOD approaches (`ACET`, `ODIN`, `OE`) degrade ID calibration of the parent model. On the contrary, ID calibration approaches (`ISOTONIC`, `SIGMOID`) perform poorly compared to that of `KDN` in the OOD region. `KDN` achieves a compromise between ID and OOD performance while having reduced confidence on wrongly classified ID samples (see MMC* in Table 1). See Appendix Table 2 for traditionally used statistics for OOD detection.

## 4  Discussion

### 4.1  Related Works and Our Contributions

Recently, Liu et al. (2023) showed that a deep network aware of distance (distance from training points) is able to capture the uncertainty associated with its prediction better than traditional approaches. However, the concept of distance awareness has previously been used by a number of approaches in the literature to detect OOD points. These approaches attempt to learn a generative model in the projected latent space and control the likelihoods far away from the training data. For example, Ren et al. (2019) used the likelihood ratio test to detect OOD samples. Wan et al. (2018) modified the training loss so that the downstream projected features follow a Gaussian distribution. In contrast to the above approaches, Liu et al. (2023) proposed an in-training approach which learns a map from the input space to a latent space and trains a Gaussian process model on top of that. Distance awareness from training points improves OOD detection and the Gaussian process improves ID calibration. However, OOD detection works as long as there are two distinguishable score sets for ID and OOD points, whereas calibration aims at estimating the true predictive uncertainty of these points (see Appendix A for details). In our work, we propose a simple post-hoc calibration approach using geodesic distance and achieve the aforementioned distance awareness from the training data by replacing the affine function over the nearest polytope with a Gaussian kernel. The proposed approach works on both random forests and deep networks. We validated our proposed approach on both tabular and vision datatsets, whereas most of the existing methods are tailor-made for vision problems.

### 4.2  Conclusions

In this paper, we have demonstrated a simple intuition that transforms traditional deep discriminative models into a type of binning and kerneling approach. The bin boundaries are determined by the internal structure learned by the parent approach, and the Geodesic distance encodes the low-dimensional structure learned by the model. The vision experiments in this paper show how the model can be applied to deeper networks by employing a front-end encoder to extract local image features (as explained in the first paragraph of Section 3.2). Instead of applying KDN to the entire `ViT`, we applied it solely to the final fully connected layers with ReLU activations. This reduces the effective depth of the model, ensuring that KDN can be applied even to encoders that do not rely on ReLU activations. Moreover, the geodesic distance introduced in this paper may have a wider impact on understanding the internal structure of the deep discriminative models, which may be of interest for future work. However, a key limitation of our approach is the need to store parameters for each populated polytope, which may hinder scalability to very large datasets. One potential solution is to merge nearby polytopes into a single representative polytope. Our code, including the package and the approach proposed in this manuscript, is available from `https://github.com/neurodata/kdg`. Appendix Section A answers a number of frequently asked questions about this work.

## Acknowledgements

The authors thank the support of the NSF-Simons Research Collaborations on the Mathematical and Scientific Foundations of Deep Learning (NSF grant 2031985) and THEORINET. This work was graciously

supported by the Defense Advanced Research Projects Agency (DARPA) Lifelong Learning Machines program through contracts FA8650-18-2-7834 and HR0011-18-2-0025. Research was partially supported by funding from Microsoft Research and the Kavli Neuroscience Discovery Institute. We also thank the helpful discussion with Will LeVine, Tyler M. Tomita, Ali Geisa, Tiffany Chu and Jacob Desman.

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

## A   Frequently Asked Questions

We received questions while presenting the proposed approach in different venues, and as we wish to reserve the main text for the core contents, we have compiled the explanation to these questions here.

**What is the relation between OOD detection and OOD confidence calibration?**   Our work addresses both in- and out-of-distribution (ID and OOD) calibration. While traditional ID calibration methods like Isotonic and Sigmoid regression focus on achieving calibrated inference within the ID region, they do not address OOD calibration. Conversely, there is another group of literature which is primarily concerned about detecting OOD points. These approaches such as ACET, OE, and ODIN mainly focus on OOD detection rather than OOD calibration. Calibration is harder than detection, akin to how regression is harder than classification. To see that calibration is harder than detection, consider the fact that a well-calibrated model can perform detection, but a model capable of detecting OOD points may not be calibrated. OOD detection works as long as there are two distinguishable score sets for ID and OOD points, whereas calibration aims at estimating the true predictive uncertainty of these points. To our knowledge, only Meinke et al. (2021) explicitly addressed OOD calibration (Section 3 Theorem 1 in their paper), but they do not consider ID calibration. Our work treats calibration problems as a continuum between the ID and OOD regions rather than addressing them separately.

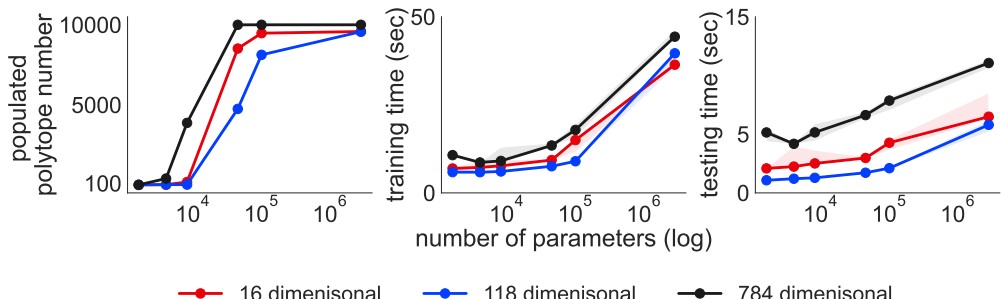

Figure 6: **Scaling plots for KDN on datasets with different feature dimensions.** The training sample size is fixed at 10,000 and the number of nodes in each layer is increased in a four layer fully connected network. Total number of polytopes populated by the training data saturates to training size (left) while training (middle) as well as testing time (right) scales linearly with node number. With over-parameterized networks, which is the norm, they typically exhibit this property that the number of populated polytopes is equal to the sample size.

**Isn't this just another out-of-distribution detection approach?** One may argue that the experimental settings essentially reduce the task to a binary one: whether to output normal confidence for an ID sample or just a prior for an OOD sample. A binary OOD detector can also immediately output a confidence based on the prior if a sample is identified as OOD. However, without ID calibration, there is no reason to expect that confidence to be calibrated within the ID region. In the experimental setting, as one moves further away from the ID setting, an OOD calibrated classifier will output the prior. However, close to the ID setting, the OOD calibrated classifier will output a probability that the sample is in any given class. This is in contrast to an OOD detector, which, regardless of how close or far the data are from the ID data, if is OOD, it always effectively outputs the prior. We would say that our method subsumes OOD detection, because it also includes ID calibration, and OOD calibration. To our knowledge, there are no other papers demonstrating any algorithm with all these properties.

**Isn't this just another kernel density estimation (KDE) approach?** There are similarities between KDE and the proposed method. However, there is an indicator function in Equation 7 which is implemented using the geodesic distance proposed in the draft (which is absent in KDE), which makes KDN, KDF scale better with higher dimensions. Moreover, the center and bandwidth of the Gaussian kernels are estimated in a data-driven manner using the representation learned by the parent discriminative approach.

**Using Gaussian kernel for posterior estimation is not novel. Isn't this similar to Gaussian Process regression?** Our proposed approach utilizes the internal structure learned by the parent approach to build an estimate of the posterior and scale for really high dimensional datasets. However, it is folk's knowledge that Gaussian Process does not scale beyond dimension 20.

**Number of polytopes grows exponentially with the size of network. Isn't the approach computationally infeasible?** Many people were concerned about the runtime of our approach, possibly it could be an exponential function of the number of nodes. However, we did additional experiments (see Fig. 6) where we show training and testing time both are linear in the number of the nodes. Moreover, training time is only 200 seconds even when there are 40,000 nodes running on a MacBook Pro with an Apple M1 Max chip and 64 GB of RAM. The number of total polytopes in KDN is upper bounded by the training sample size as we only consider the polytopes populated by training data (see the first paragraph of Section 2.3 and Equation 7). With overparameterized networks, which is the norm, they typically exhibit this property as shown in Fig. 6 that the number of populated polytopes is equal to the sample size

**How does the proposed approach perform in term of runtime compared to the baseline approaches?** OOD calibration approaches such as ACET, OE and ODIN take about 2 days, an hour, 6 hours, respectively on GPUs. In-distribution calibration methods such as isotonic regression and sigmoid regression

take a few minutes and use CPUs. Our approach addresses both ID and OOD calibration while taking a few minutes to train on CPUs, rather than GPUs. All the computations were performed for producing the results in Table 1 using a MacBook Pro with an Apple M1 Max chip and 64 GB of RAM.

**Why is there no non-asymptotic performance guarantee?** Because of "No Free Lunch" theorem, it is impossible to provide a non-asymptotic guarantee that works for all training distributions. Hence, we provided extensive empirical experiments encompassing simulation, tabular and vision datasets to build intuition about the finite sample performance of the proposed approach.

**Why did you use a new statistic called OCE for OOD calibration measure?** OCE measures OOD calibration according to Equation 4 and assumes that true class conditional priors for the datasets are known. On the contrary, ECE is used for measuring ID calibration and is a surrogate measure used when true posteriors are not known. For example, in Figure 3 where we know the true distribution, we have used Hellinger distance from the true posteriors instead of ECE. Note that measuring ECE requires accuracy which requires class labels and there is no class label available for the OOD samples in our setting.

**Why is there no OOD baseline approach for the tabular benchmark datasets?** `ACET`, `ODIN`, `OE` are tailor-made for vision problems, therefore we can not run them on tabular data using the author provided codes. To the best of our knowledge, the tabular OOD method (Ren et al., 2019) that we found does OOD detection, not calibration. As it does not yield any posterior, we could not benchmark with the above method.

**In Table 1, it seems OOD approaches including ACET and ODIN can't even beat the parent model under OOD settings.** `ACET` and `ODIN` highly depend on the model architecture and the nature of the ID and OOD testsets. Moreover, they also depend on the OOD set used to train them. We used the OOD set used by the authors of the above algorithms. All these factors contribute to their inconsistent performance across datasets and model architecture. Tajwar et al. (2021) found a qualitatively similar result.

**What would be the AUROC, FPR, i.e. metrics commonly used in OOD detection?** To save space in the main text, we have added AUROC and FPR in Appendix Table 2. Note that these scores are used for OOD detection. However, we are addressing both ID and OOD calibration (Equation 4 in the main text). OOD detection works as long as there are two distinguishable score sets for ID and OOD points, whereas calibration aims at estimating the true predictive uncertainty of these points, both in ID and OOD regions. That being said, a well-calibrated model can perform detection, but a model capable of detecting OOD points may not be calibrated.

Table 2: `KDN` **has nearly similar or better AUROC and FPR to those of baseline OOD detection approaches** '↑' and '↓' indicate whether higher and lower values are better, respectively.

| | Dataset | Statistics | Parent | KDN | ISOTONIC | SIGMOID | ACET | ODIN | OE | Focal | Dual Focal |
|---|---|---|---|---|---|---|---|---|---|---|---|
| ID | CIFAR-10 | Accuracy(%) ↑ | $98.06 \pm 0.00$ | $97.45 \pm 0.00$ | $98.16 \pm 0.00$ | $98.10 \pm 0.00$ | $\mathbf{98.23 \pm 0.00}$ | $97.97 \pm 0.00$ | $97.94 \pm 0.00$ | $94.79 \pm 0.00$ | $95.23 \pm 0.00$ |
| OOD | CIFAR-100 | AUROC ↑ | $0.88 \pm 0.00$ | $0.95 \pm 0.01$ | $\mathbf{0.96 \pm 0.00}$ | $0.92 \pm 0.01$ | $\mathbf{0.96 \pm 0.00}$ | $0.92 \pm 0.01$ | $0.92 \pm 0.01$ | $0.93 \pm 0.00$ | $0.93 \pm 0.00$ |
| | | FPR@95 ↓ | $0.13 \pm 0.01$ | $0.12 \pm 0.01$ | $0.14 \pm 0.01$ | $0.14 \pm 0.01$ | $0.12 \pm 0.00$ | $0.34 \pm 0.02$ | $\mathbf{0.10 \pm 0.01}$ | $0.37 \pm 0.01$ | $0.39 \pm 0.01$ |
| | SVHN | AUROC ↑ | $0.88 \pm 0.01$ | $0.97 \pm 0.00$ | $0.89 \pm 0.00$ | $0.90 \pm 0.00$ | $0.98 \pm 0.00$ | $0.94 \pm 0.02$ | $\mathbf{0.99 \pm 0.00}$ | $0.93 \pm 0.02$ | $0.94 \pm 0.01$ |
| | | FPR@95 ↓ | $0.10 \pm 0.02$ | $0.08 \pm 0.01$ | $0.15 \pm 0.00$ | $0.12 \pm 0.00$ | $0.05 \pm 0.01$ | $0.17 \pm 0.03$ | $\mathbf{0.01 \pm 0.00}$ | $0.44 \pm 0.09$ | $0.45 \pm 0.07$ |
| | Noise | AUROC ↑ | $0.85 \pm 0.00$ | $\mathbf{0.98 \pm 0.01}$ | $0.82 \pm 0.00$ | $0.89 \pm 0.00$ | $\mathbf{0.99 \pm 0.00}$ | $\mathbf{0.99 \pm 0.00}$ | $\mathbf{0.99 \pm 0.01}$ | $0.89 \pm 0.06$ | $0.86 \pm 0.03$ |
| | | FPR@95 ↓ | $0.17 \pm 0.00$ | $0.03 \pm 0.03$ | $0.12 \pm 0.00$ | $0.11 \pm 0.01$ | $\mathbf{0.00 \pm 0.00}$ | $\mathbf{0.00 \pm 0.00}$ | $\mathbf{0.00 \pm 0.01}$ | $0.78 \pm 0.31$ | $0.99 \pm 0.00$ |
| ID | CIFAR-100 | Accuracy(%) ↑ | $\mathbf{86.72 \pm 0.00}$ | $85.46 \pm 0.00$ | $85.33 \pm 0.00$ | $86.61 \pm 0.00$ | $85.07 \pm 0.01$ | $86.56 \pm 0.00$ | $86.09 \pm 0.00$ | $79.37 \pm 0.06$ | $80.81 \pm 0.00$ |
| OOD | CIFAR-10 | AUROC ↑ | $0.89 \pm 0.01$ | $\mathbf{0.97 \pm 0.00}$ | $0.83 \pm 0.02$ | $0.90 \pm 0.01$ | $0.88 \pm 0.02$ | $0.90 \pm 0.01$ | $0.90 \pm 0.01$ | $0.81 \pm 0.00$ | $0.81 \pm 0.01$ |
| | | FPR@95 ↓ | $0.41 \pm 0.02$ | $\mathbf{0.05 \pm 0.01}$ | $0.61 \pm 0.04$ | $0.38 \pm 0.03$ | $0.57 \pm 0.04$ | $0.58 \pm 0.02$ | $0.50 \pm 0.03$ | $0.66 \pm 0.01$ | $0.68 \pm 0.01$ |
| | SVHN | AUROC ↑ | $0.91 \pm 0.01$ | $\mathbf{0.98 \pm 0.00}$ | $0.87 \pm 0.01$ | $0.91 \pm 0.01$ | $0.88 \pm 0.02$ | $0.81 \pm 0.02$ | $0.92 \pm 0.01$ | $0.89 \pm 0.02$ | $0.89 \pm 0.02$ |
| | | FPR@95 ↓ | $0.39 \pm 0.03$ | $\mathbf{0.04 \pm 0.01}$ | $0.59 \pm 0.00$ | $0.38 \pm 0.03$ | $0.51 \pm 0.06$ | $0.62 \pm 0.05$ | $0.34 \pm 0.03$ | $0.89 \pm 0.02$ | $0.51 \pm 0.06$ |
| | Noise | AUROC ↑ | $0.98 \pm 0.01$ | $\mathbf{1.00 \pm 0.00}$ | $0.93 \pm 0.01$ | $0.98 \pm 0.01$ | $\mathbf{1.00 \pm 0.00}$ | $0.98 \pm 0.01$ | $0.98 \pm 0.02$ | $0.99 \pm 0.01$ | $\mathbf{1.00 \pm 0.00}$ |
| | | FPR@95 ↓ | $0.06 \pm 0.05$ | $\mathbf{0.00 \pm 0.00}$ | $0.42 \pm 0.03$ | $0.07 \pm 0.06$ | $0.01 \pm 0.00$ | $0.08 \pm 0.04$ | $0.15 \pm 0.17$ | $\mathbf{0.00 \pm 0.00}$ | $\mathbf{0.00 \pm 0.00}$ |
| ID | SVHN | Accuracy(%) ↑ | $93.52 \pm 0.01$ | $92.84 \pm 0.00$ | $93.64 \pm 0.01$ | $93.65 \pm 0.01$ | $93.22 \pm 0.01$ | $93.52 \pm 0.01$ | $\mathbf{93.76 \pm 0.01}$ | $60.39 \pm 0.00$ | $61.02 \pm 0.00$ |
| OOD | CIFAR-10 | AUROC ↑ | $0.95 \pm 0.01$ | $0.97 \pm 0.01$ | $0.96 \pm 0.01$ | $0.94 \pm 0.01$ | $0.95 \pm 0.01$ | $0.96 \pm 0.01$ | $\mathbf{0.99 \pm 0.00}$ | $0.52 \pm 0.01$ | $0.54 \pm 0.01$ |
| | | FPR@95 ↓ | $0.30 \pm 0.07$ | $0.06 \pm 0.04$ | $0.25 \pm 0.08$ | $0.34 \pm 0.09$ | $0.38 \pm 0.06$ | $0.18 \pm 0.07$ | $\mathbf{0.02 \pm 0.01}$ | $0.97 \pm 0.00$ | $0.97 \pm 0.00$ |
| | CIFAR-100 | AUROC ↑ | $0.95 \pm 0.01$ | $\mathbf{0.97 \pm 0.01}$ | $0.96 \pm 0.01$ | $0.95 \pm 0.01$ | $0.95 \pm 0.01$ | $0.95 \pm 0.01$ | $0.95 \pm 0.01$ | $0.52 \pm 0.01$ | $0.55 \pm 0.00$ |
| | | FPR@95 ↓ | $0.30 \pm 0.07$ | $\mathbf{0.06 \pm 0.02}$ | $0.25 \pm 0.07$ | $0.35 \pm 0.08$ | $0.37 \pm 0.05$ | $0.19 \pm 0.06$ | $\mathbf{0.04 \pm 0.02}$ | $0.97 \pm 0.00$ | $0.97 \pm 0.00$ |
| | Noise | AUROC ↑ | $0.96 \pm 0.02$ | $0.98 \pm 0.03$ | $0.98 \pm 0.01$ | $0.96 \pm 0.01$ | $\mathbf{1.00 \pm 0.00}$ | $0.99 \pm 0.00$ | $\mathbf{1.00 \pm 0.00}$ | $0.63 \pm 0.06$ | $0.64 \pm 0.06$ |
| | | FPR@95 ↓ | $0.24 \pm 0.16$ | $0.04 \pm 0.05$ | $0.16 \pm 0.11$ | $0.26 \pm 0.13$ | $\mathbf{0.00 \pm 0.00}$ | $0.01 \pm 0.01$ | $0.01 \pm 0.01$ | $1.0 \pm 0.00$ | $1.0 \pm 0.00$ |

**Why did you choose CIFAR-10, CIFAR-100 and SVHN for your vision experiments?** We have done vision experiments using CIFAR-10 (10 classes), CIFAR-100 (100 classes) and SVHN (10 classes and bigger training size) as ID datasets. We emphasize that CIFAR-10, CIFAR-100 and SVHN are some of the hardest ID and OOD pairs according to various papers (Nalisnick et al., 2018; Fort et al., 2021), and hence they are adopted as the benchmarking datasets by many of the papers in the literature. Doing experiments with extremely large datasets like imagenet (14 million images) is computationally and storage-wise expensive using our current implementation. We will pursue extremely large datasets in future. Many relevant papers on OOD calibration use only small- and mid-sized datasets (Gardner et al., 2024; Borisov et al., 2022; Ulmer et al., 2020). We acknowledge this limitation in the paper.

**As shown in Algorithm 2, in each iteration, Geodesic kernel values between the current training input and the entire dataset need to be computed, each of which involves comparing all possible activation paths through the neural network (Equation 12 and Algorithm 1). And Algorithm 2 iterates over every single training instance. This will become extremely challenging on modern neural networks with miliions, or even billions, of parameters (nodes) trained on large-scale datasets consisting of millions of training samples.** We address this scalability issue with two solutions:

1. **Parallelization for Efficient Kernel Computation**: Our kernel is computed as the product of Hamming distances between activation patterns (binary strings of '1' and '0') at each layer for two samples. Since these calculations are independent across layers and sample pairs, they can be efficiently parallelized. We leverage the *cdist* function from *scipy.spatial.distance*, which enables fast pairwise distance computations. This parallel implementation reduced runtime from days to just minutes.

2. **Efficient Computation for Large Models**: For extremely large models, such as vision transformers (ViTs) in Section 3.2.2, we employ a front-end encoder to extract local image features (as described in Section 3.2, first paragraph). Instead of applying KDN to the entire ViT, we fit it only to the final fully connected layers. This reduces the size of the model by about 90% while maintaining performance, allowing more efficient KDN computation.

**As dimension continues to increase, densities of a Gaussian kernel tend to concentrate on a thin shell, where densities decay exponentially both going outward (away from the centre) and inward (towards the centre). Thus, a low class conditional density estimated by the Gaussian kernel does not necessarily imply the inference sample is OOD. Such high-dimensional feature space (large ) is commonly encountered in practical tasks.** In such high dimensional spaces, most of the probability mass concentrates on a thin shell at a certain radius from the mean due to the rapid growth of volume on the surface of the shell. While the density of a Gaussian distribution is highest at the mean and decays exponentially with Euclidean distance from the mean, the total probability mass at a given radius is given by the integral of the probability density over that region. The volume of the surface at that radius increases rapidly with higher dimensions (Fraser, 1984). In high dimensions, this results in the counterintuitive effect where the peak probability mass is not at the mean but at some distance from it. However, this phenomenon does not affect our approach. As shown in Equation 5, we evaluate the Gaussian kernel at a single point x, not over an entire surface or volume. Since the kernel density is maximized at the center and decays exponentially outward, the volume concentration effect does not alter the inference process in our method.

## B Proofs

### B.1 Proof of Proposition 1

To prove that $d$ is a valid distance metric for $\mathcal{P}_n$, we need to prove the following four statements. Here we use the notation $d(r, s)$ to denote the distance between two samples $\boldsymbol{x}_i \in Q_r$ and $\boldsymbol{x}_s \in Q_s$.

1. $d(r, s) = 0$ when $r = s$.
   **Proof:** By definition, $\mathcal{K}(r, s) = 1$ and $d(r, s) = 0$ when $r = s$.

2. $d(r, s) > 0$ when $r \neq s$.
   **Proof:** By definition, $0 \leq \mathcal{K}(r, s) < 1$ and $d(r, s) > 0$ for $r \neq s$.

3. $d$ is symmetric, i.e., $d(r, s) = d(s, r)$.
   **Proof:** By definition, $\mathcal{K}(r, s) = \mathcal{K}(s, r)$ which implies $d(r, s) = d(s, r)$.

4. $d$ follows the triangle inequality, i.e., for any three polytopes $Q_r, Q_s, Q_t \in \mathcal{P}_n$: $d(r, t) \leq d(r, s) + d(s, t)$.
   **Proof:** Let $\mathcal{A}_r$ denote the sequence of elements in the activation mode for a particular polytope $r$. $B$ is the cardinality of $\mathcal{A}_r$. Below $c(\cdot)$ denotes the cardinality of the sequence. We can write:

$$B \geq c((\mathcal{A}_r \cap \mathcal{A}_s) \cup (\mathcal{A}_s \cap \mathcal{A}_t)) \tag{26}$$
$$= c(\mathcal{A}_r \cap \mathcal{A}_s) + c(\mathcal{A}_s \cap \mathcal{A}_t) - c(\mathcal{A}_r \cap \mathcal{A}_s \cap \mathcal{A}_t)$$
$$\geq c(\mathcal{A}_r \cap \mathcal{A}_s) + c(\mathcal{A}_s \cap \mathcal{A}_t) - c(\mathcal{A}_r \cap \mathcal{A}_t).$$

Rearranging the above equation, we get:

$$B - c(\mathcal{A}_r \cap \mathcal{A}_t) \leq B - c(\mathcal{A}_r \cap \mathcal{A}_s) + B - c(\mathcal{A}_s \cap \mathcal{A}_t)$$
$$\implies 1 - \frac{c(\mathcal{A}_r \cap \mathcal{A}_t)}{B} \leq 1 - \frac{c(\mathcal{A}_r \cap \mathcal{A}_s)}{B} + 1$$
$$- \frac{c(\mathcal{A}_s \cap \mathcal{A}_t)}{B}$$
$$\implies d(r, t) \leq d(r, s) + d(s, t). \tag{27}$$

## B.2 Proof of Proposition 2

Note that first we find the nearest polytope to the inference point $x$ using Geodesic distance and use Gaussian kernel locally for $x$ within that polytope. Here the Gaussian kernel uses Euclidean distance from the kernel center to $x$ (within the numerator of the exponent). The value out of the Gaussian kernel decays exponentially with the increasing distance of the inference point from the kernel center. We first expand $\hat{g}_y(\mathbf{x})$:

$$\hat{g}_y(\mathbf{x}) = \frac{\hat{f}_y(\mathbf{x})\hat{P}_Y(y)}{\sum_{k=1}^K \hat{f}_k(x)\hat{P}_Y(k)}$$
$$= \frac{\tilde{f}_y(\mathbf{x})\hat{P}_Y(y) + \frac{b}{\log(n)}\hat{P}_Y(y)}{\sum_{k=1}^K (\hat{f}_k(\mathbf{x})\hat{P}_Y(k) + \frac{b}{\log(n)}\hat{P}_Y(k))}$$

As the inference point $\mathbf{x}$ becomes more distant from training samples (and more distant from all of the Gaussian centers), we have that $\mathcal{G}(\mathbf{x}, \hat{\mu}_r, \hat{\Sigma}_r)$ becomes smaller. Thus, $\forall y, \tilde{f}_y(\mathbf{x})$ shrinks. More formally, $\forall y$,

$$\lim_{d_\mathbf{x} \to \infty} \tilde{f}_y(\mathbf{x}) = 0.$$

We can use this result to then examine the limiting behavior of our posteriors as the inference point $\mathbf{x}$ becomes more distant from the training data:

$$\lim_{d_\mathbf{x} \to \infty} \hat{g}_y(\mathbf{x}) = \lim_{d_\mathbf{x} \to \infty} \frac{\tilde{f}_y(\mathbf{x})\hat{P}_Y(y) + \frac{b}{\log(n)}\hat{P}_Y(y)}{\sum_{k=1}^K (\tilde{f}_k(\mathbf{x})\hat{P}_Y(k) + \frac{b}{\log(n)}\hat{P}_Y(k))}$$
$$= \frac{\hat{P}_Y(y)}{\sum_{k=1}^K \hat{P}_Y(k)}$$
$$= \hat{P}_Y(y).$$

Note that without the constant term $\frac{b}{\log(n)}$, the class priors in the numerator will be 0 and hence we will not have Proposition 2.

## C   Hardware and Software Configurations

- Operating System: Linux (ubuntu 20.04), macOS (Ventura 13.2.1)

- VM Size: Azure Standard D96as v4 (96 vcpus, 384 GiB memory)

- GPU: Apple M1 Max

- Software: Python 3.8, scikit-learn $\geq$ 0.22.0, tensorflow-macos$\leq$2.9, tensorflow-metal $\leq$ 0.5.0.

## D   Simulations

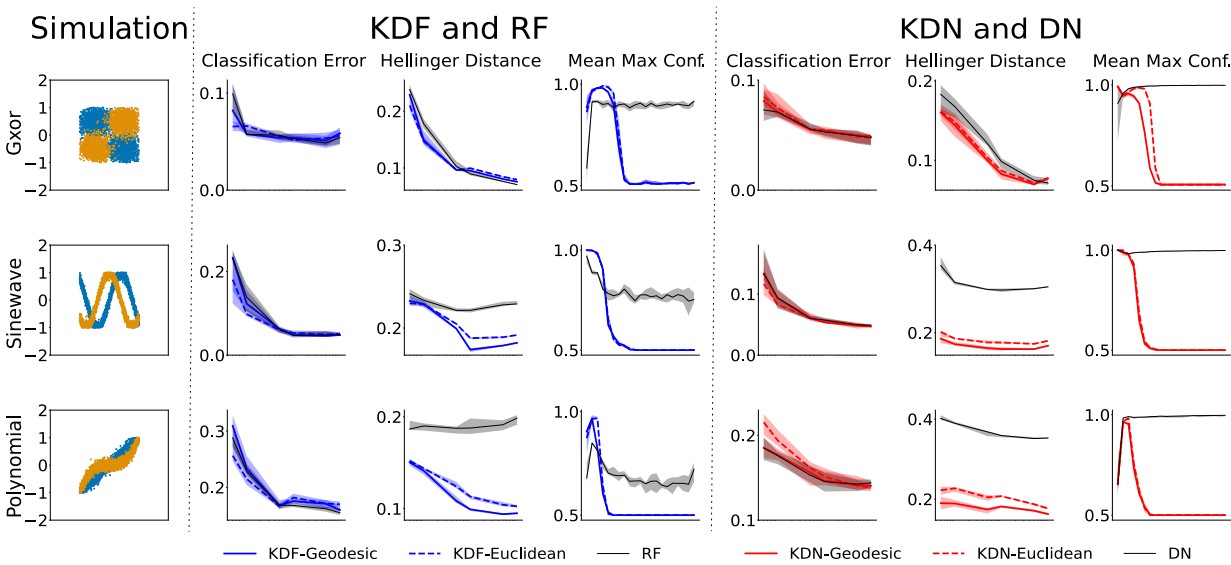

Figure 7: **Additional simulation result.** `KDF` and `KDN` improve both in- and out-of-distribution calibration of their respective parent algorithms while maintaining nearly similar classification accuracy on the simulation datasets.

We construct six types of binary class simulations:

- *Spiral* is a two-class classification problem with the following data distributions: let $K$ be the number of classes and $S \sim \text{multinomial}(\frac{1}{K}\vec{1}_K, n)$. Conditioned on $S$, each feature vector is parameterized by two variables, the radius $r$ and an angle $\theta$. For each sample, $r$ is sampled uniformly in $[0, 1]$. Conditioned on a particular class, the angles are evenly spaced between $\frac{4\pi(k-1)t_K}{K}$ and $\frac{4\pi(k)t_K}{K}$, where $t_K$ controls the number of turns in the spiral. To inject noise along the spirals, we add Gaussian noise to the evenly spaced angles $\theta' : \theta = \theta' + \mathcal{N}(0, 0.09)$. The observed feature vector is then $(r\cos(\theta), r\sin(\theta))$.

- *Circle* is a two-class classification problem with equal class priors. Conditioned on being in class 0, a sample is drawn from a circle centered at $(0, 0)$ with a radius of $r = 0.75$. Conditioned on being in class 1, a sample is drawn from a circle centered at $(0, 0)$ with a radius of $r = 1$, which is cut off by the region bounds. To inject noise along the circles, we add Gaussian noise to the circle radii $r' : r = r' + \mathcal{N}(0, 0.01)$.

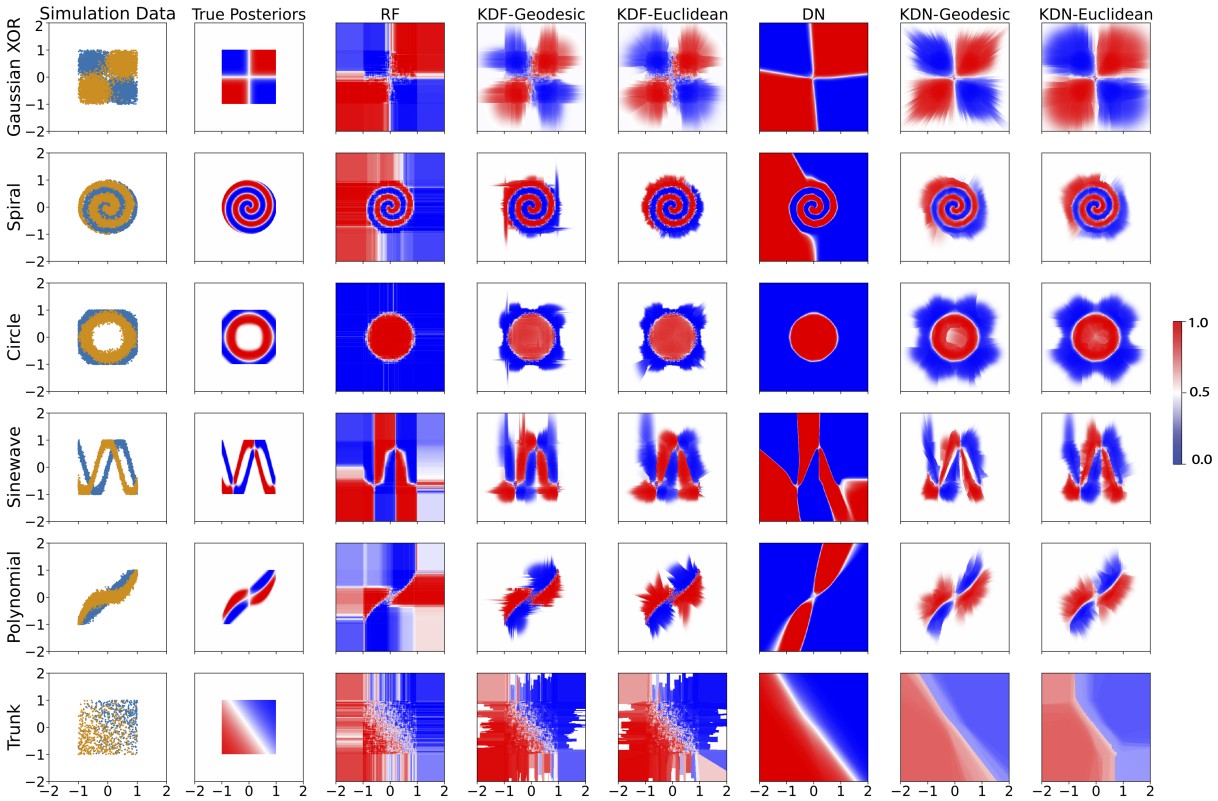

Figure 8: **Visualization of true and estimated posteriors for class 0 from five binary class simulation experiments.** *Column 1*: 10,000 training points with 5,000 samples per class sampled from 6 different simulation setups for binary class classification. Trunk simulation is shown for two dimensional case. The class labels are indicated by yellow and blue colors. *Column 2-8*: True and estimated class conditional posteriors from different approaches. The posteriors estimated from KDN and KDF are better calibrated for both in- and out-of-distribution regions compared to those of their parent algorithms.

- *Trunk* is a two-class classification problem with gradually increasing dimension and equal class priors. The class conditional probabilities are Gaussian:

$$P(X|Y = 0) = \mathcal{G}(\mu_1, I),$$
$$P(X|Y = 1) = \mathcal{G}(\mu_2, I),$$

  where $\mu_1 = \mu, \mu_2 = -\mu$, $\mu$ is a $d$ dimensional vector whose $i$-th component is $(\frac{1}{i})^{1/2}$ and $I$ is $d$ dimensional identity matrix.

- *Sinewave* is a two-class classification problem based on sine waves. Conditioned on being in class 0, a sample is drawn from the distribution $y = \cos(\pi x)$. Conditioned on being in class 1, a sample is drawn from the distribution $y = \sin(\pi x)$. We inject Gaussian noise to the sine wave heights $y' : y = y' + \mathcal{N}(0, 0.01)$.

- *Gaussian XOR* is a two-class classification problem with equal class priors. Conditioned on being in class 0, a sample is drawn from a mixture of two Gaussians with means $\pm[0.5, -0.5]^\top$ and standard deviations of 0.25. Conditioned on being in class 1, a sample is drawn from a mixture of two Gaussians with means $\pm[0.5, -0.5]^\top$ and standard deviations of 0.25.

- *Polynomial* is a two-class classification problem with the following data distributions: $y = x^a$. Conditioned on being in class 0, a sample is drawn from the distribution $y = x^1$. Conditioned on

being in class 1, a sample is drawn from the distribution $y = x^3$. Gaussian noise is added to variables $y' : y = y' + \mathcal{N}(0, 0.01)$.

Table 3: Hyperparameters for `RF` and `KDF`.

| Hyperparameters | Value |
|---|---|
| n_estimators | 500 |
| max_depth | $\infty$ |
| min_samples_leaf | 1 |
| $\lambda$ | $1 \times 10^{-6}$ |
| $b$ | $\exp\left(-10^{-7}\right)$ |

Table 4: Hyperparameters for `ReLU`-net and `KDN` on Tabular data.

| Hyperparameters | Value |
|---|---|
| number of hidden layers | 4 |
| nodes per hidden layer | 1000 |
| optimizer | Adam |
| learning rate | $3 \times 10^{-4}$ |
| $\lambda$ | $1 \times 10^{-6}$ |
| $b$ | $\exp\left(-10^{-7}\right)$ |

## E   Extended Results on OpenML-CC18 data suite

See Figure 9, 10, 11 and 12 for extended results on OpenML-CC18 data suite.

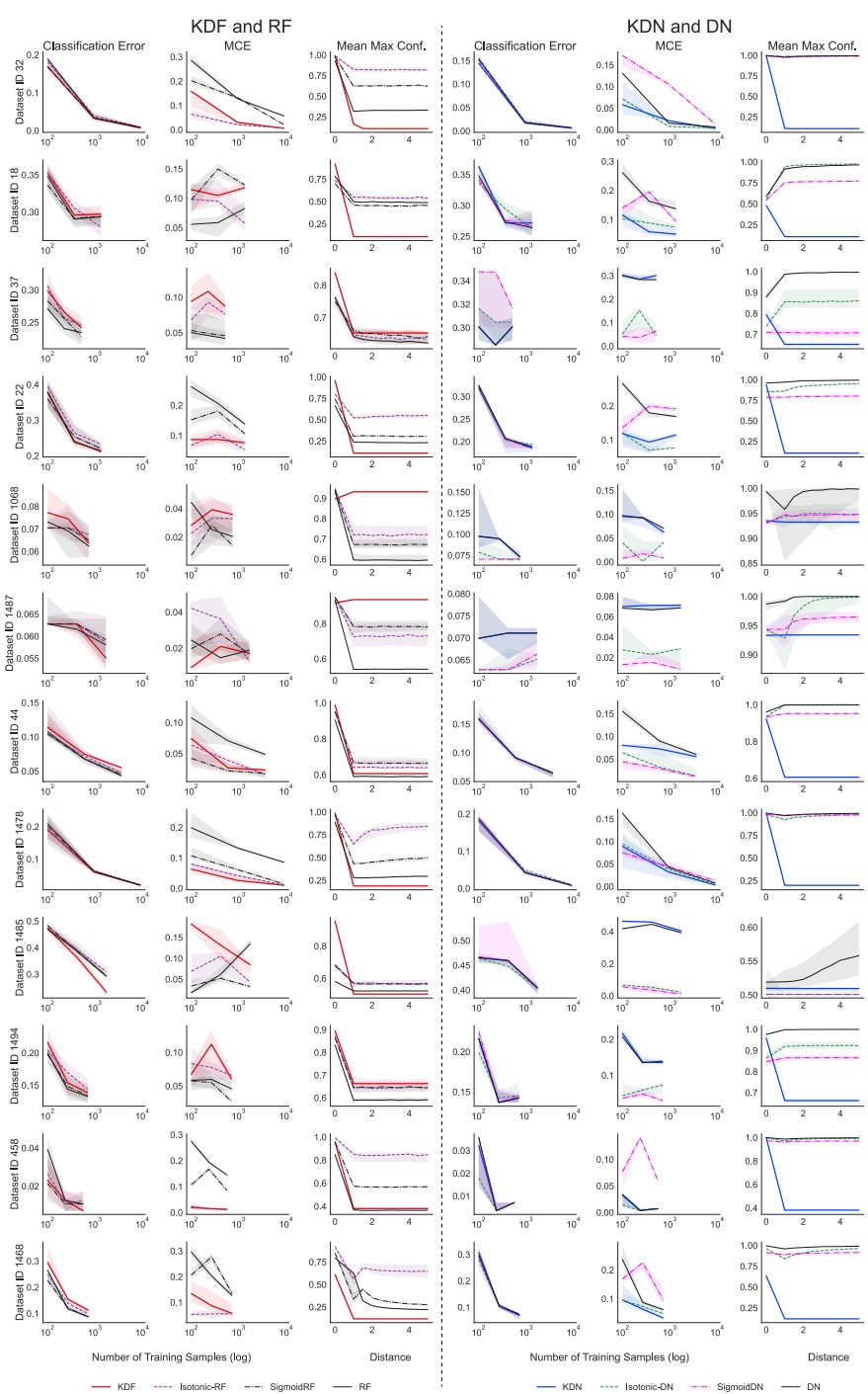

Figure 9: **Extended results on OpenML-CC18 datasets.** *Left:* Performance (classification error, ECE and mean max confidence) of KDF on different Openml-CC18 datasets. *Right:* Performance (classification error, ECE and mean max confidence) of KDN on different Openml-CC18 datasets.

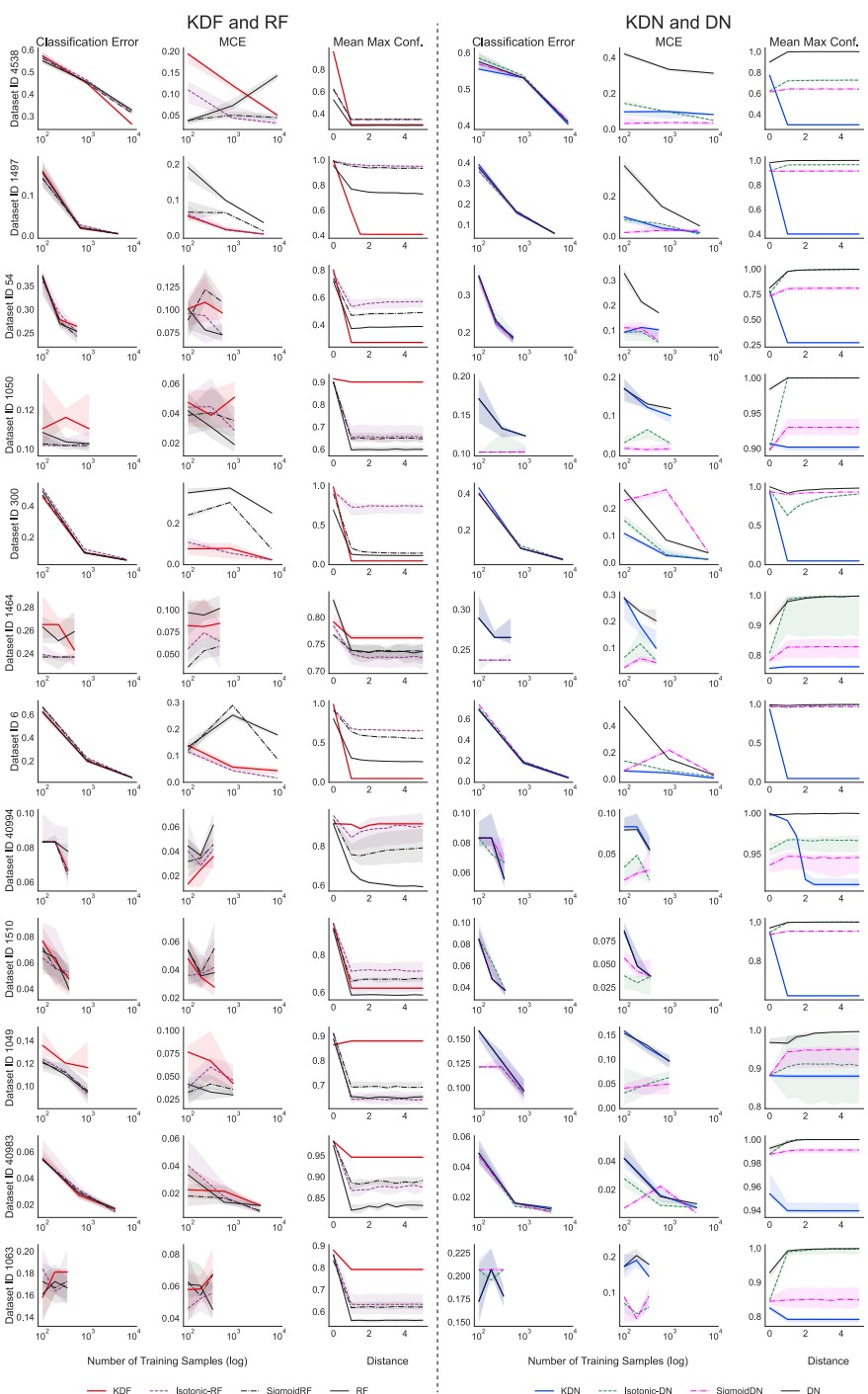

Figure 10: **Extended results on OpenML-CC18 datasets (continued).** *Left:* Performance (classification error, ECE and mean max confidence) of KDF on different Openml-CC18 datasets. *Right:* Performance (classification error, ECE and mean max confidence) of KDN on different Openml-CC18 datasets.

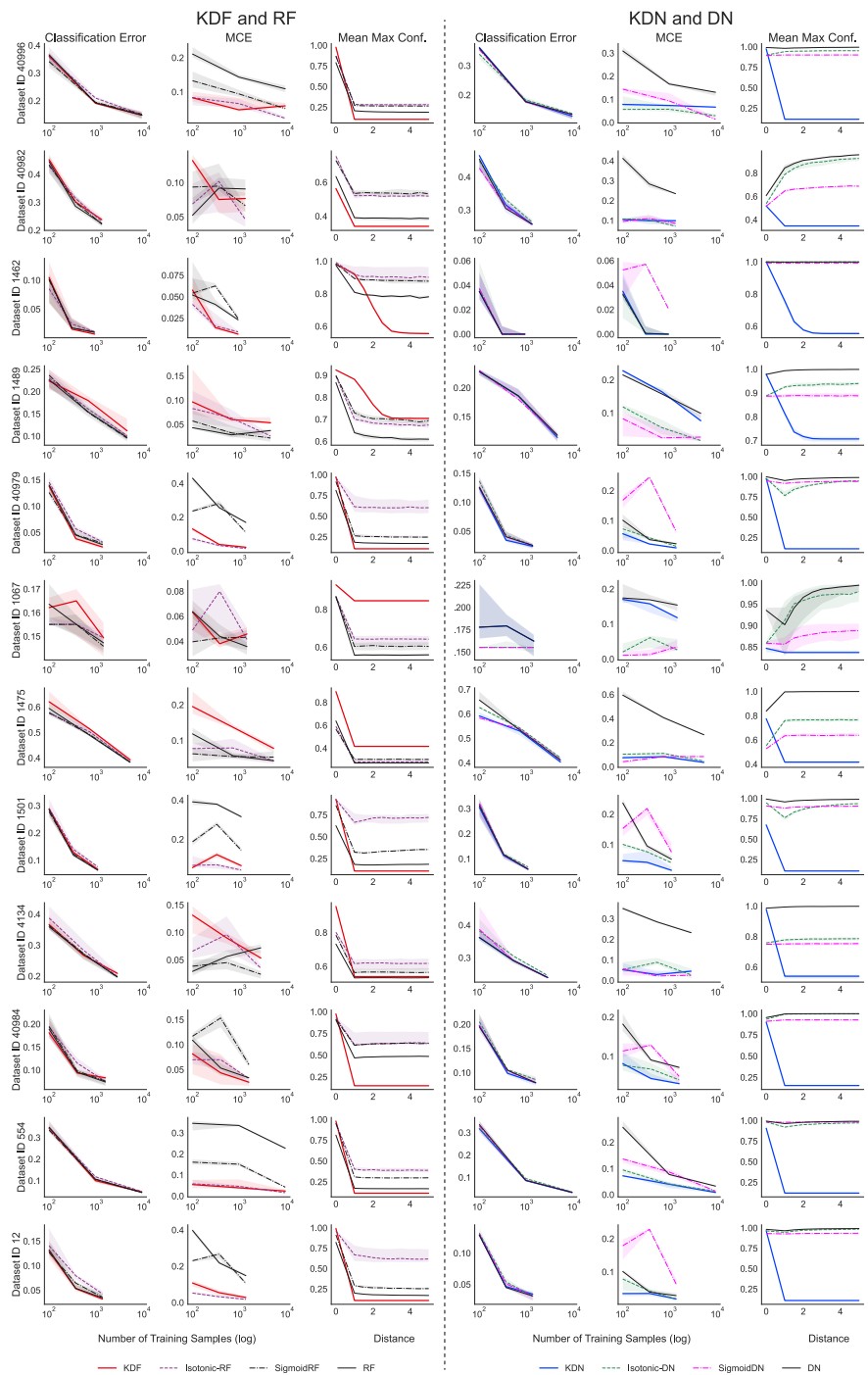

Figure 11: **Extended results on OpenML-CC18 datasets (continued).** *Left:* Performance (classification error, ECE and mean max confidence) of KDF on different Openml-CC18 datasets. *Right:* Performance (classification error, ECE and mean max confidence) of KDN on different Openml-CC18 datasets.

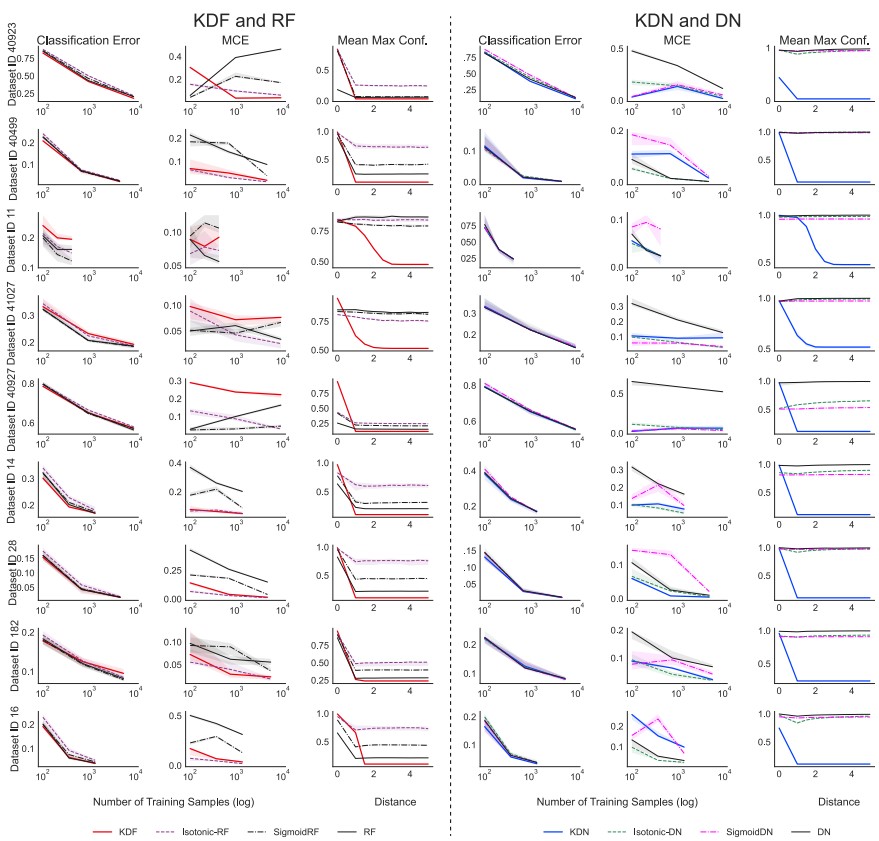

Figure 12: **Extended results on OpenML-CC18 datasets (continued).** *Left:* Performance (classification error, ECE and mean max confidence) of KDF on different Openml-CC18 datasets. *Right:* Performance (classification error, ECE and mean max confidence) of KDN on different Openml-CC18 datasets.

