# OpenReview forum: "Simple Calibration via Geodesic Kernels"
_TMLR — Accepted by TMLR_

### Review · Reviewer_F1TG · 2025-03-12

**Summary Of Contributions:**

This paper studies calibration for out-of-distribution (OOD) data. The proposed method models the class-conditional data distribution (or confidence) of each class as a sum of Gaussian kernels, similar to a Gaussian mixture model but with some pruning. The model resembles a model in the form of a union of polytopes, such as decision forests and ReLU networks. These models provide the class probabilities for each sample, which can then be used to compute the confidence of the classifier.

**Audience:**

Yes

**Claims And Evidence:**

Yes

**Requested Changes:**

1. Add a discussion on whether the kernel in Eqn. (10) is still meaningful when the neural network is very deep.
2. The presentation of the experiment results can be improved. Figure 2 is not very parsable. I am not sure how it shows that the proposed method improves OOD calibration. Section 3.2.2 needs a major edit. I don't think it is a good idea to use complete noise from a uniform distribution as OOD. For example, you can use SVHN as OOD for the CIFAR dataset.

**Strengths And Weaknesses:**

Strengths:
1. The OOD calibration problem is an interesting problem that is worth studying.
2. The description of the method is quite clear.
3. The reported experiment results show that the method has better OOD calibration than classical methods such as Platt scaling.

Weaknesses:
1. The application of the proposed method is very narrow. The paper only discusses decision forests and ReLU networks. And even for ReLU networks, the kernel defined in Eqn. (10) in Section 2.4.2 is very sparse when the network is very deep, because almost no two sample will have the exact same activation path. Although the experiments in Section 3.2.2 consider ViT, I cannot understand this section so I am not sure if the method can be applied to a more realistic setting.
2. The proposed method seems a bit crude. It is essentially approximating the class-conditional data distribution with a Gaussian mixture like model. I don't think this is applicable to complex data. In the experiments, only Section 3.2.2 uses CIFAR, but I cannot understand the result of this section.
3. For a ViT B16, how is the network kernel defined? Do you strictly the definition in Eqn. (10)? Wouldn't that be a very sparse kernel? And why do you use complete complete white noise as OOD? Wouldn't that make the task too easy? I am not convinced by the results in Section 3.2.2.

---

> ### Author Response · Authors · 2025-04-07
> **Response**
>
> Thank you for your thoughtful comments. We believe we have addressed all your concerns in our responses below.
> > The application of the proposed method is very narrow. The paper only discusses decision forests and ReLU networks. And even for ReLU networks, the kernel defined in Eqn. (10) in Section 2.4.2 is very sparse when the network is very deep, because almost no two samples will have the exact same activation path. Although the experiments in Section 3.2.2 consider ViT, I cannot understand this section so I am not sure if the method can be applied to a more realistic setting.
>
> > Add a discussion on whether the kernel in Eqn. (10) is still meaningful when the neural network is very deep.
>
>  Thank you for your insightful feedback! We agree that the sparsity of the kernel in Eqn. (10) (now Eqn. 12) can become a concern in very deep networks, where the activation paths for samples with the same class label may diverge significantly. To address this issue, we conducted an empirical experiment by training a fully-connected network with increasing layers and nodes (see Section 2.5). In this experiment, we measured the kernel similarity between samples of the same and different class labels.
> Our findings indicate that, when the network weights are initialized randomly, the partitions become random, and the activation patterns of samples are highly disjoint. This leads to a low kernel similarity for very deep networks. However, as the network learns the classification task, the weights are adjusted in such a way that the activation patterns of samples belonging to the same class become more aligned. This suggests that, although the kernel may initially appear sparse in deep networks with random weights, it becomes more informative once the network starts learning from the data, with the similarity between samples of the same class increasing. Consequently, the kernel remains meaningful in the context of a learned network, even when it is very deep.
> Moreover, in practice, machine learning practitioners do not typically use fully-connected networks with such extreme depth due to the challenges associated with gradient propagation.
>
> For large models, such as Vision Transformers (ViTs) discussed in Section 3.2.2, we employ a front-end encoder to extract local image features (as explained in the first paragraph of Section 3.2). Instead of applying KDN to the entire ViT, we apply it solely to the final fully connected layers with ReLU activations. This reduces the effective depth of the model, addressing the sparsity issue and ensuring that KDN can be applied even to encoders that do not rely on ReLU activations.
>
> > The presentation of the experiment results can be improved. Figure 2 is not very parsable. I am not sure how it shows that the proposed method improves OOD calibration.
>
>  We are grateful to the reviewer for raising this point. Indeed, we missed the x-axis labels for the low-dimensional simulations in Figure 2A, which made the figure unclear. We apologize for this oversight. We have corrected the figure by adding the necessary labels and revised the caption to provide a more detailed explanation of the results. This should help in clarifying how the proposed method improves OOD calibration. Thank you for pointing this out!
>
>
> > The proposed method seems a bit crude. It is essentially approximating the class-conditional data distribution with a Gaussian mixture like model. I don't think this is applicable to complex data.
>
>  We appreciate your acknowledgment of the simplicity of our approach, which is also reflected in the paper's title. However, our method is not a Gaussian mixture model; rather, for a given inference point, it considers only the single nearest Gaussian in terms of Geodesic distance. We have clarified this after Equation 7. The effectiveness of our approach arises from the combination of the exponential decay property of Gaussians and the use of Geodesic distance, which leverages the internal structure learned by the parent model.
> Regarding applicability to complex datasets, our approach is first validated on simulated datasets with known ground truth. We further evaluated it on diverse real-world datasets, including 45 OpenML tabular datasets (Figure 4, Appendix Figures 9–12) and vision datasets (CIFAR-10, CIFAR-100, and SVHN as in-distribution, with others as OOD, as shown in Table 1). These results demonstrate the robustness of our method across various settings.

---

> > ### Author Response · Authors · 2025-04-07
> > **continued response**
> >
> > > Section 3.2.2 needs a major edit. I don't think it is a good idea to use complete noise from a uniform distribution as OOD. For example, you can use SVHN as OOD for the CIFAR dataset.
> >
> >  We have improved Section 3.2.2 by providing a more detailed discussion. For the vision transformer, we utilize the transformer as a front-end encoder and apply the KDN model to the last two fully connected layers. We described the intuition behind this approach in the first paragraph of Section 3.2 and have now explicitly referred back to it in Section 3.2.2 for ease of access.
> > Regarding OOD selection, we use noise primarily as a sanity check, where the inference point is semantically far from the training data. Interestingly, even in this simple setup, most baseline algorithms struggle due to a fundamental limitation of the ReLU activation. This phenomenon is described in Theorem 3.1 of [1]. However, we also considered near-OOD cases, such as CIFAR-100 and SVHN, when using CIFAR-10 as the training set in Table 1.
> >
> > **References:**
> > [1] Hein, Matthias, Maksym Andriushchenko, and Julian Bitterwolf. "Why relu networks yield high-confidence predictions far away from the training data and how to mitigate the problem." Proceedings of the IEEE/CVF conference on computer vision and pattern recognition. 2019.
> >
> > Please let us know if you have any additional concerns.

---

### Review · Reviewer_psMA · 2025-03-22

**Summary Of Contributions:**

The submitted manuscript proposes a kernel-based method to address the model calibration issue for the out-of-distribution (OOD) detection task while achieving calibration on in-distribution data. Following the existing literature that interprets the deep neural network as partitioning the feature space into a union of polytopes, the proposed method uses a Geodesic Kernel to estimate the similarity between the partitioning paths of different training/inference data samples. The estimated similarity facilitates the computation of soft membership of samples to each polytope as well as the fractions of class-specific data being sorted into each polytope. Furthermore, within each polytope, a Gaussian Kernel with learnable parameters is introduced to decay the class conditional density exponentially with respect to the distance to the centre of the kernel. Finally, Following the Bayes' rule, both the estimated class conditional density and the fractions, along with a prior class probability, are used to estimate the posterior class probability of a given inference point, which is subsequently used for OOD detection and OOD calibration evaluation. This work also proposes a metric to evaluate the calibration performance for OOD detection.

**Audience:**

Yes

**Broader Impact Concerns:**

N.A.

**Claims And Evidence:**

Yes

**Requested Changes:**

### **Requested Clarifications**

1. The proposed OOD calibration error needs to have established connection to the definition Expected Calibration Error in order to be meaningful.

2. Solutions to both scalability issues are required.

3. Comparisons with latest model calibration and OOD detection methods are preferred.

**Strengths And Weaknesses:**

### **Strengths**

1. The proposed method is intuitive and well defined.
1. The paper is nicely written with well-defined notations.
2. The work brings attention to an important feature (calibration) for DNNs designed for a popular task - OOD detection.


### **Weanesses**
1. The proposed OOD Calibration Error is unconvincing. It lacks a connection to the definition of Expected Calibration Error.

2. The proposed OOD Calibration Error (OCE) (Equation 18) is tailored specifically for the proposed method. However, an empirical prior class distribution is not a gold standard in identifying the OOD samples. Detection of OOD data can vary across the methods. Among the compared OOD detection methods, ODIN thresholds the top prediction probability, which does not force the OOD samples to have a prior class probability; and both ACET and OE (also threshold based) encourage OOD samples to have a uniform class distribution. Such an evaluation is biased against the compared OOD methods.

3. Scalability issue #1 - computational resources. As shown in Algorithm 2, in each iteration, Geodesic kernel values between the current training input and the entire dataset need to be computed, each of which involves comparing all possible activation paths through the neural network (Equation 10 and Algorithm 1). Note that there are
> $B = \prod_{l=1}^L |\mathcal{N}_l|,$
> where $L$ is the total number of layers and $\mathcal{N}_l$ is the set of nodes corresponding to the l$^{\text{th}}$ layer.

    possible activation paths for each training input. And Algorithm 2 iterates over every single training instance. This will become extremely challenging on modern neural networks with miliions, or even billions, of parameters (nodes) trained on large-scale datasets consisting of millions of training samples.

4. Scalability issue #2 - Behaviour of Gaussian kernel changes in high-dimensional space. As $d$ continues to increase, densities of a Gaussian kernel tend to concentrate on a thin shell, where densities decay exponentially both going outward (away from the centre) and inward (towards the centre). Thus, a low class conditional density estimated by the Gaussian kernel does not necessarily imply the inference sample is OOD. Such high-dimensional feature space (large $d$) is commonly encountered in practical tasks.

5. Comparison methods. The compared model calibration methods, *Isotonic Regression (IR)* and *Sigmoid (Platt) Regression (SR)*, are relatively old and under-performing in contemporary model calibration research. It is also known that post-processing based calibration methods, including IR, SR, temperature scaling etc., are particularly vulnerable to calibration on OOD data or OOD detections. On the contrary, training-based model calibration methods typically have much better performances on OOD data. For example, the following works (non-exhaustive) have demonstrated their calibration effectivenesses on OOD data. However, all of the training-based model calibration methods are left out of the comparison.
    1. Mukhoti, J., Kulharia, V., Sanyal, A., Golodetz, S., Torr, P., & Dokania, P. (2020). Calibrating deep neural networks using focal loss. Advances in neural information processing systems, 33, 15288-15299.
    2. Tao, L., Dong, M., & Xu, C. (2023, July). Dual focal loss for calibration. In International Conference on Machine Learning (pp. 33833-33849). PMLR.
    3. Bohdal, O., Yang, Y., & Hospedales, T. M. (2023). Meta-Calibration: Learning of Model Calibration Using Differentiable Expected Calibration Error. Transactions on Machine Learning Research, 1-21.
    4. Yu, Y., Bates, S., Ma, Y., & Jordan, M. (2022). Robust calibration with multi-domain temperature scaling. Advances in Neural Information Processing Systems, 35, 27510-27523.

6. The experiments are relatively small-scale.

---

> ### Author Response · Authors · 2025-04-07
> **Response**
>
> We thank the reviewer for the intuitive comments. We believe we have addressed all your concerns in the response below.
>
> > The proposed OOD calibration error needs to have established connection to the definition Expected Calibration Error in order to be meaningful.
>
> > “The proposed OOD Calibration Error is unconvincing. It lacks a connection to the definition of Expected Calibration Error.
> The proposed OOD Calibration Error (OCE) (Equation 18) is tailored specifically for the proposed method. However, an empirical prior class distribution is not a gold standard in identifying the OOD samples. Detection of OOD data can vary across the methods. Among the compared OOD detection methods, ODIN thresholds the top prediction probability, which does not force the OOD samples to have a prior class probability; and both ACET and OE (also threshold based) encourage OOD samples to have a uniform class distribution. Such an evaluation is biased against the compared OOD methods.”
>
>  Thank you for raising this foundational question. We previously failed to provide the justification for this statistic, and we agree that an OOD calibration metric must have principled grounding to be meaningful. Below, we provide the conceptual and mathematical rationale for OCE. Expected Calibration Error (ECE) quantifies the difference between model confidence and accuracy by binning inference points. However, ECE is applicable only to in-distribution (ID) data, as accuracy requires ground truth labels. Therefore, ECE cannot be directly applied to OOD points. Our proposed OOD Calibration Error (OCE) is motivated by the goal outlined in Equation 4, serving as an analogous measure for OOD regions. While ECE assesses calibration in the ID region, OCE evaluates calibration in the OOD region.
>
> To provide intuition, consider Bayes' formula (Equation 1):
>
> $$P_{Y|X}(y|\mathbf{x}) = \frac{P_{X|Y}(\mathbf{x}|y) P_{Y}(y)}{\sum_{k=1}^K P_{X|Y}(\mathbf{x}|k) P_Y(k)}, \quad \forall y \in \mathcal{Y}$$
>
> In OOD regions, where no training data exists, $P_{X|Y}$ should be equal across all classes, as the model lacks evidence to favor any particular class $y$ for an inference point $x$. This is the motivation behind adding a constant in Equation 8. Substituting this into Bayes' formula, for data that is out of distribution, yields:
>
> $$P_{Y|X}(y|\mathbf{x}) = P_Y(y).$$
> This aligns with Proposition 2 (Appendix B.2) and intuitively implies that when faced with an unknown sample, the model’s prediction should default to the majority class. For instance, if the classifier has seen more dog images than cat images, it should predict an unknown image as a dog with higher probability. We have added texts to Section 2.1 to describe the above intuition better.
> Many OOD detection methods assume uniform class priors (see Figure 1 and Proposition 2.2 in [7], which assume equal priors for binary classification). To ensure fairness, we use equal priors in all vision experiments. Notably, Outlier Exposure performs second-best among the baselines under this setup. Additionally, we report conventional OOD detection metrics in Appendix Table 2 for completeness.
>
> > Solutions to both scalability issues are required.
>
> > “Scalability issue #1 - computational resources. As shown in Algorithm 2, in each iteration, Geodesic kernel values between the current training input and the entire dataset need to be computed, each of which involves comparing all possible activation paths through the neural network (Equation 10 and Algorithm 1). Note that where  is the total number of layers and  is the set of nodes ....”
>
>  We have added this comment to our FAQ section. We address this scalability issue with two solutions:
> - **Parallelization for Efficient Kernel Computation:**
>
>  Our kernel is computed as the product of Hamming distances between activation patterns (binary strings of ‘1’ and ‘0’) at each layer for two samples. Since these calculations are independent across layers and sample pairs, they can be efficiently parallelized. We leverage the *cdist* function from *scipy.spatial.distance*, which enables fast pairwise distance computations. This parallel implementation reduced runtime from days to just minutes. We will make the code open-source after the acceptance of the paper.
>
>
> - **Efficient Computation for Large Models:**
>
>  For extremely large models, such as vision transformers (ViTs) in Section 3.2.2, we employ a front-end encoder to extract local image features (as described in Section 3.2, first paragraph). Instead of applying KDN to the entire ViT, we fit it only on the final fully connected layers. This reduces the model size by about $95$% while maintaining performance, allowing for more efficient KDN computation.

---

> ### Author Response · Authors · 2025-04-07
> **continued response**
>
> > Scalability issue #2 - Behaviour of Gaussian kernel changes in high-dimensional space. As  continues to increase, densities of a Gaussian kernel tend to concentrate on a thin shell, where densities decay exponentially both going outward (away from the centre) and inward (towards the centre). Thus, a low class conditional density estimated by the Gaussian kernel does not necessarily imply the inference sample is OOD. Such high-dimensional feature space (large) is commonly encountered in practical tasks.
>
> Good observation! This insight deepened our understanding of the proposed approach. We have added this comment to our FAQ section. The unintuitive behavior of Gaussian distributions in high-dimensional spaces is well-documented (e.g., [1,2]). In such high dimensional spaces, most of the probability mass concentrates on a thin shell at a certain radius from the mean due to the rapid growth of volume on the surface of the shell.
> While the density of a Gaussian distribution is still highest at the mean and decays exponentially with Euclidean distance from the mean, the total probability mass at a given radius is given by the integral of the probability density over that region. The volume of the surface at that radius increases rapidly with higher dimensions (shown in [1,2]). In high dimensions, this results in the counterintuitive effect where the peak probability mass is not at the mean but at some distance from it.
> However, this phenomenon does not affect our approach. As shown in Equation 7, we evaluate the Gaussian kernel at a single point x, not over an entire surface or volume. Since the kernel density is maximized at the center and decays exponentially outward, the volume concentration effect does not alter the inference process in our method.
> Thus, while volume effects challenge probability mass estimation in high-dim spaces, our approach—evaluating fixed-point Gaussian kernels—remains unaffected, since the inference depends only on the point-wise kernel value, not integrated volume.
>
> > Comparisons with latest model calibration and OOD detection methods are preferred.
> “Comparison methods. The compared model calibration methods, Isotonic Regression (IR) and Sigmoid (Platt) Regression (SR), are relatively old and under-performing in contemporary model calibration research. It is also known that post-processing based calibration methods, including IR, SR, temperature scaling etc., are particularly vulnerable to calibration on OOD data or OOD detections. On the contrary, training-based model calibration methods typically have much better performances on OOD data. For example, the following works (non-exhaustive) have demonstrated their calibration effectivenesses on OOD data. However, all of the training-based model calibration methods are left out of the comparison.”
>
> We appreciate the reviewer providing specific baselines to add to the paper. Notably, our approach is a post-hoc calibration method, and now, we have now added in-training calibration baselines from [3] and [4] to Table 1 as suggested by the reviewer.
> However, we encountered challenges reproducing some methods:
> - The authors of [5] did not provide code in their paper.
>
> - The provided code link in [6] (https://github.com/yaodongyu/MDTS) is broken.
>
> From our experiments, training-based calibration methods in [3] and [4] achieve good ID and OOD calibration but at the cost of reduced in-distribution classification accuracy. KDN still performs best overall.
>
> **References**:
> 1.  https://andrewcharlesjones.github.io/journal/high-dim-gaussians.html?hl=en-US
> 2. Fraser, Marshall. "The grazing goat in n dimensions." The Two-Year College Mathematics Journal 15.2 (1984): 126-134
> 3. Mukhoti, J., Kulharia, V., Sanyal, A., Golodetz, S., Torr, P., & Dokania, P. (2020). Calibrating deep neural networks using focal loss. Advances in neural information processing systems, 33, 15288-15299.
> 4. Tao, L., Dong, M., & Xu, C. (2023, July). Dual focal loss for calibration. In International Conference on Machine Learning (pp. 33833-33849). PMLR.
> 5. Bohdal, O., Yang, Y., & Hospedales, T. M. (2023). Meta-Calibration: Learning of Model Calibration Using Differentiable Expected Calibration Error. Transactions on Machine Learning Research, 1-21.
> 6. Yu, Y., Bates, S., Ma, Y., & Jordan, M. (2022). Robust calibration with multi-domain temperature scaling. Advances in Neural Information Processing Systems, 35, 27510-27523.
> 7. Kristiadi, Agustinus, Matthias Hein, and Philipp Hennig. "Being bayesian, even just a bit, fixes overconfidence in relu networks." International conference on machine learning. PMLR, 2020.
> 8. Tajwar, Fahim, et al. "No true state-of-the-art? ood detection methods are inconsistent across datasets." arXiv preprint arXiv:2109.05554 (2021).
>
> Please let us know if you have any additional concerns.

---

### Review · Reviewer_CZ96 · 2025-03-23

**Summary Of Contributions:**

The paper addresses the task of simultaneously calibrating models for both in-distribution (ID) and out-of-distribution (OOD) instances. Typically, approaches focus exclusively on either ID or OOD calibration. The proposed method clusters training data based on the activations of the model. At test time, the method locates the closest polytope center in input space and then applies a Gaussian kernel, whose parameters are determined during the fitting of the KDX model. The effectiveness of the proposed approach is validated through classification
experiments conducted on simulated, tabular, and vision datasets. Experimental results demonstrate strong performance in terms of both ID and OOD calibration, as well as OOD detection.

**Audience:**

Yes

**Broader Impact Concerns:**

There are no ethical concerns that would require more detailed attention in the paper.

**Claims And Evidence:**

Yes

**Requested Changes:**

The questions stated above in the section on Strengths and Weaknesses need answers in the paper.

**Strengths And Weaknesses:**

Strengths:

* Method achieves good performance in terms of calibration and OOD detection.

* Method can be applied to existing networks, given that it uses ReLU activations.

Weaknesses and questions:

* From Figure 5 we can see that the number of polytopes goes to 10 000 quickly, which is also the size of the training data. This raises the question of whether fitting the kdx model is even necessary? If all the polytopes contain only a single point, then it is the same as finding the
nearest neighbour. Did the authors also consider finding the nearest neighbour and calculating the gaussian kernel to that one?

* In Figure 5, assessing scaling behavior is challenging because the models quickly reach the maximum number of polytopes (equal to the size of the training data) using relatively few nodes. It would be more interesting to see this experiment with a larger training set

* The class conditional priors were fixed for OCE calculation. It remains unclear how the class conditional priors were fixed.

* Since we can see confidence intervals for the results, it would be nice to mention how many times the experiments were repeated.

* Looking at the formula (5), we see that during the inference, the value of f_y(x) depends on the test set size. Is there a big difference, if we look at the test set as a whole or each test point individually?

---

> ### Author Response · Authors · 2025-04-07
> **Response**
>
> We sincerely appreciate your thoughtful comments. Below, we address your concerns.
>
> > From Figure 5 we can see that the number of polytopes goes to 10,000 quickly, which is also the size of the training data. This raises the question of whether fitting the kdx model is even necessary? If all the polytopes contain only a single point, then it is the same as finding the nearest neighbour.
>
> Reply: Our approach is not equivalent to nearest neighbor search. In KDX, we explicitly prune unpopulated polytopes and only retain those that are populated by training data. More importantly, distances between polytopes are not measured using Euclidean distance but rather geodesic distance—defined using the internal structure learned by the parent model. This structure-aware distance captures meaningful relationships in the data that simple nearest neighbor methods do not.
>
> > Did the authors also consider finding the nearest neighbour and calculating the gaussian kernel to that one?
>
> Reply: Yes, we considered this approach in Section 3.1.1. It is the Euclidean nearest neighbor approach. However, as shown in Figure 2, using the nearest polytope center in Euclidean space leads to scalability issues in high-dimensional settings and results in poor calibration. Our method, which leverages geodesic distance, provides better calibration and scales more effectively as dimensionality increases.
>
> > In Figure 5, assessing scaling behavior is challenging because the models quickly reach the maximum number of polytopes (equal to the size of the training data) using relatively few nodes. It would be more interesting to see this experiment with a larger training set.
>
> Reply: To provide clearer insights, we have revised Figure 5 (now Figure 6) using fewer nodes, preventing immediate saturation of the populated polytopes to the training set size and plotting the x-axis in log-scale. As established in [1], the number of polytopes in a ReLU network grows exponentially with network depth and width. However, since we consider only polytopes populated by training samples, their count quickly stabilizes at the training set size. With overparameterized networks, which is the norm, they typically exhibit this property that the number of populated polytpoes is equal to the sample size.
>
> > The class conditional priors were fixed for OCE calculation. It remains unclear how the class conditional priors were fixed.
>
> Reply: In our tabular data experiments, training samples are drawn according to empirical class priors in the overall dataset. For vision experiments, we assume uniform class priors when sampling training points. We have clarified this in Section 2.9.
>
>
> > Since we can see confidence intervals for the results, it would be nice to mention how many times the experiments were repeated.
>
> Reply: Thank you for this suggestion! We have now explicitly stated the number of experiment repetitions in each relevant section (marked in red for easy reference).
>
> > Looking at the formula (5), we see that during the inference, the value of f_y(x) depends on the test set size. Is there a big difference, if we look at the test set as a whole or each test point individually?
>
> Reply: Excellent observation! To clarify, $n_y$​ represents the total number of training samples in class y, while $n_{ry}$ is the number of training samples from class y within polytope $Q_r$​. Both values depend exclusively on the training set, meaning that $f_y(x)$ is independent of the test set size. We have revised the text following Equation (5) (now Equation 7) to make this explicit.
>
>
> Reference:
> [1] Montufar, G. F., Pascanu, R., Cho, K., & Bengio, Y. (2014). On the number of linear regions of deep neural networks. Advances in neural information processing systems, 27.
>
>
> Please let us know if you have any additional concerns.

---

> > ### Comment · Action_Editor_d5HV · 2025-05-21
> >
> > Dear Reviewer CZ96,
> >
> > Please review the authors’ response in conjunction with the comments from other reviewers and provide your final recommendation.
> >
> > Best,
> >
> > Your Action Editors

---

> ### Comment · Action_Editor_d5HV · 2025-04-27
>
> Dear Reviewer CZ96,
>
> Thank you for your efforts in reviewing this submission. Kindly review the authors’ response and the comments from other reviewers, and submit your final recommendation.
>
> Best,
>
> Your Action Editors

---

### Decision · Action_Editor_d5HV · 2025-06-03

**Recommendation:** Accept with minor revision

**Comment:**

This submission received reviews from three experts, all of whom provided constructive comments and valuable suggestions. Following the discussion, the overall feedback is positive. However, several points need to be addressed in the camera-ready version:

- “With overparameterized networks, which is the norm, they typically exhibit this property that the number of populated polytopes is equal to the sample size.” Reviewer CZ96 suggested that this point should be clearly stated in the paper.
- I concur with the Reviewers F1TG  and psMA that the experiments are conducted on small-scale datasets using relatively simple models. It would strengthen the paper to include a discussion on how the proposed method could generalize to more complex datasets and deeper neural networks.

**Audience:**

Researchers focused on confidence calibration would likely find this submission relevant.

**Claims And Evidence:**

The main problem is how to calibrate model confidence on both in-distribution (ID) and out-of-distribution (OOD) data. The idea builds on how deep networks divide the feature space into many flat regions, or polytopes. A geodesic distance is then used to measure how far these regions are from each other, and a Gaussian kernel helps to separate samples based on this distance. This method works to some extent, but the evidence so far comes mostly from small datasets and simple models.